# Approximating Two-Layer Feedforward Networks for Efficient Transformers

**Róbert Csordás**[1]   **Kazuki Irie**[2†]   **Jürgen Schmidhuber**[1,3]

[1]The Swiss AI Lab IDSIA, USI & SUPSI   [2]Harvard University   [3]AI Initiative, KAUST
{robert,juergen}@idsia.ch, kirie@fas.harvard.edu

## Abstract

How to reduce compute and memory requirements of neural networks (NNs) without sacrificing performance? Many recent works use sparse Mixtures of Experts (MoEs) to build resource-efficient large language models (LMs). Here we introduce several novel perspectives on MoEs, presenting a general framework that *unifies* various methods to *approximate two-layer NNs* (e.g., feedforward blocks of Transformers), including product-key memories (PKMs). Leveraging insights from this framework, we propose methods to improve both MoEs and PKMs. Unlike prior work that compares MoEs with dense baselines under the *compute-equal* condition, our evaluation condition is *parameter-equal*, which is crucial to properly evaluate LMs. We show that our MoEs are competitive with the *dense* Transformer-XL on both the WikiText-103 and enwik8 datasets at two different scales, while being much more resource-efficient. This demonstrates that MoEs are relevant not only to extremely large LMs but also to any-scale resource-efficient LMs. Our code is public.[1]

## 1 Introduction

Despite impressive results recently achieved by large language models (LLMs; Radford et al. (2019); Brown et al. (2020); Rae et al. (2021)), vast resource requirement remains their obvious limitation. In fact, most existing LLMs, such as GPT-3 (Brown et al., 2020), cannot be trained, fine-tuned or even evaluated without access to enormous compute. Many recent works strive to develop LLMs that, at least, enable inference with limited resources (e.g., on consumer hardware), e.g., by building "smaller" yet capable LMs (Touvron et al., 2023; Taori et al., 2023; Chiang et al., 2023) or developing post-training quantization methods (Zafrir et al., 2019; Dettmers et al., 2022). While these

---

† Work done at IDSIA.

[1]https://github.com/robertcsordas/moe

methods are gaining popularity, a principled solution for resource-efficient neural networks (NNs) remains elusive.

One promising approach explored by several recent works on extremely-large LMs is the sparse mixture of experts (MoE; Shazeer et al. (2017); Lewis et al. (2021); Lepikhin et al. (2021); Fedus et al. (2022); Clark et al. (2022); Chi et al. (2022)). Unlike their *dense* counterparts, MoEs only compute a subset of their activations (i.e, only a few *experts*) at each step, offering reduced computation and memory costs. However, MoEs are not yet generally adopted as a generic/to-go approach, perhaps because of certain common beliefs on MoEs: (1) They are hard to train (involving complex engineering tricks to prevent collapsing), (2) they are not competitive against their dense counterparts with the *same number of parameters* (in fact, prior work focuses on FLOP-equal comparison, "unfairly" comparing MoEs against dense baselines with many fewer trainable parameters), and finally, (3) they are reserved for extremely large models (they are rarely/never considered to further improve the efficiency of "small" models). Indeed, even prior works on MoE-based Transformer LMs only deploy MoEs in a few feedforward blocks; while ideally, *all* such blocks should benefit from replacement by MoEs. Here we challenge these common beliefs, and propose novel perspectives on MoEs.

We present MoEs within a unified framework of methods that approximate two-layer feedforward networks, which includes product-key memories (PKMs; Lample et al. (2019)) and top-$k$ sparsification. This principled view not only allows us to conceptually group and compare MoEs with PKMs, it also provides insights on design choices for improving these methods. Our resulting MoE Transformer variant outperforms our improved PKMs, and performs as well as or even outperforms the dense baseline, while using a fraction of its compute for both training

and inference. Importantly, unlike prior work, we compare our MoEs with dense baselines with the same number of total trainable parameters, which is crucial for proper evaluation in language modeling. We conduct experiments on the standard WikiText-103 (at two different model scales) and Enwik8 datasets. We demonstrate that MoEs are not limited to extremely-large LMs, but useful as a generic approach for resource-efficient NNs at any scale, and in line with the recent trend of improving "smaller" models (Touvron et al., 2023; Taori et al., 2023; Chiang et al., 2023). Finally, we release a CUDA kernel for our MoE layers which allows for achieving faster wall clock time and large memory reduction compared to the dense model.[2]

## 2 Background

Transformers (Vaswani et al., 2017) have two main building blocks: the self-attention layer (Parikh et al., 2016; Cheng et al., 2016; Bahdanau et al., 2015), and the two-layer feedforward, i.e, multi-layer perceptron (MLP) block. Acceleration and memory reduction of the self-attention is rather well explored (see, e.g., linear attention dating back to the unnormalised linear Transformers of 1991 (Schmidhuber, 1991; Katharopoulos et al., 2020; Choromanski et al., 2021; Schlag et al., 2021)), and very efficient implementations (Dao et al., 2022) are also available. In constrast, resource-efficient MLP blocks are still underexplored. This is our main focus, and it is of particular relevance today, as the proportion of the total parameter counts, compute and memory requirements due to MLP blocks in Transformers is increasing in ever-growing LLMs.

Let $d_{\text{model}}, d_{\text{ff}}$ denote positive integers. Each Transformer MLP block consists of one up-projection layer with a weight matrix $W_1 \in \mathbb{R}^{d_{\text{ff}} \times d_{\text{model}}}$ where typically $d_{\text{ff}} = 4 d_{\text{model}}$, and one down-projection layer with parameters $W_2 \in \mathbb{R}^{d_{\text{model}} \times d_{\text{ff}}}$ that projects it back to the original size. Non-linearity (typically ReLU) is applied between these two layers. That is, an input $x \in \mathbb{R}^{d_{\text{model}}}$ is transformed to an output $y \in \mathbb{R}^{d_{\text{model}}}$ as

$$u = \text{ReLU}\left(W_1 x\right) \qquad (1)$$
$$y = W_2 u \qquad (2)$$

where $u \in \mathbb{R}^{d_{\text{ff}}}$, and we omit biases (as well as batch and time dimensions) for simplicity.

Alternatively, this layer can be viewed as a key-value memory accessed by attention (Vaswani et al. (2017)[3],Geva et al. (2021)), where keys and values are rows and columns of weight matrices $W_1$ and $W_2$:

$$W_1 = \begin{bmatrix} — & k_1^{\mathsf{T}} & — \\ — & k_2^{\mathsf{T}} & — \\ & \vdots & \\ — & k_{d_{\text{ff}}}^{\mathsf{T}} & — \end{bmatrix} \qquad (3)$$

$$W_2 = \begin{bmatrix} | & | & & | \\ v_1 & v_2 & \dots & v_{d_{\text{ff}}} \\ | & | & & | \end{bmatrix} \qquad (4)$$

where $k_i \in \mathbb{R}^{d_{\text{model}}}, v_i \in \mathbb{R}^{d_{\text{model}}}$ for $i \in \{1, ..., d_{\text{ff}}\}$. Then, the output is computed as "attention":

$$y = \sum_{i=1}^{d_{\text{ff}}} v_i \, \text{ReLU}(k_i^{\mathsf{T}} x) = \sum_{i=1}^{d_{\text{ff}}} \alpha_i v_i \qquad (5)$$

where $\alpha_i = \text{ReLU}(k_i^{\mathsf{T}} x) \in \mathbb{R}_{\geq 0}$ are the "attention weights." Note that $\alpha_i = u[i]$ where $u[i] \in \mathbb{R}$ denotes the $i$-th component of $u \in \mathbb{R}^{d_{\text{ff}}}$ in Eq. 1. Unlike the standard self-attention, the MLP block uses a ReLU activation function (instead of the softmax) without scaling.

It has been observed that, in practice, only a few of the factors $k_i^{\mathsf{T}} x$ are positive (Li et al., 2023; Shen et al., 2023), making the first layer's output, i.e., $u$, sparse. Concretely, Shen et al. (2023) report that in a Transformer with $d_{\text{model}} = 256$ and $d_{\text{ff}} = 1024$, 10% of the channels account for 90% of the total activation mass. We confirm this trend in our own preliminary study. Fig. 1 shows the average number of non-zero units in $u$ of size $d_{\text{ff}} = 2053$ in our 47M parameter dense model trained on WikiText-103 (we refer to App. A.2 for more details). The number is below 200 for all layers. This suggests that the MLP block can be *approximated* without a significant performance loss. Note that this is also supported by the findings of Zhang et al. (2022).

## 3 Approximating 2-layer MLPs

Here we present a unified view on methods to approximate 2-layer MLPs (Sec. 2) that includes many existing methods such as MoEs (Sec. 3.3) and PKMs (Sec. 3.2).

---

[2]Our non-expert CUDA implementation still has much room for further optimization.

[3]See the appendix "Two feedforward Layers = Attention over Parameter" in their paper version "arXiv:1706.03762v3."

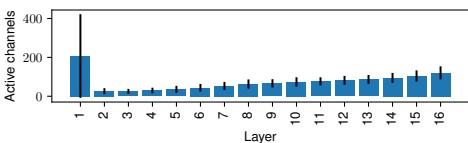

Figure 1: Number of active channels in $\boldsymbol{u}$ in our dense 47M parameter model on WikiText-103 (out of 2053 total channels). Standard deviation over all tokens of the test and validation set.

**Preliminaries.** Let $\hat{\boldsymbol{y}} \in \mathbb{R}^{d_{\text{model}}}$ denote an approximation of $\boldsymbol{y} \in \mathbb{R}^{d_{\text{model}}}$ in Eq. 5. Let $\boldsymbol{y}_i \in \mathbb{R}^{d_{\text{model}}}$ denote $\boldsymbol{y}_i = \alpha_i \boldsymbol{v}_i$ for $i \in \{1, ..., d_{\text{ff}}\}$. The core idea is to approximate the sum in Eq. 5, i.e., $\boldsymbol{y} = \sum_{i=1}^{d_{\text{ff}}} \boldsymbol{y}_i$ by only keeping a subset $\mathcal{S} \subset \{1, ..., d_{\text{ff}}\}$ of the key-value pairs, i.e., $\hat{\boldsymbol{y}} = \sum_{i \in \mathcal{S}} \boldsymbol{y}_i$. The intuition of this approximation is as follows. We assume that a good approximation $\hat{\boldsymbol{y}}$ of $\boldsymbol{y}$ is the one that minimizes their Euclidean distance $e = ||\hat{\boldsymbol{y}} - \boldsymbol{y}||_2^2 \in \mathbb{R}$, which can now be expressed as $e = ||\sum_{i \in \bar{\mathcal{S}}} \alpha_i \boldsymbol{v}_i||_2^2$ where $\bar{\mathcal{S}}$ denotes the complement of $\mathcal{S}$, i.e., $\bar{\mathcal{S}} = \{1, ..., d_{\text{ff}}\} \setminus \mathcal{S}$. Since we have $e = ||\sum_{i \in \bar{\mathcal{S}}} \alpha_i \boldsymbol{v}_i||_2^2 \leq \sum_{i \in \bar{\mathcal{S}}} \alpha_i ||\boldsymbol{v}_i||_2^2$ (triangle inequality; where the equality is achieved when $\boldsymbol{v}_i$ are orthogonal), this upper-bound $\sum_{i \in \bar{\mathcal{S}}} \alpha_i ||\boldsymbol{v}_i||_2^2$ can be minimized if each term $c_i = \alpha_i ||\boldsymbol{v}_i||_2^2 \in \mathbb{R}$ are small. If we further assume that all value vectors $\boldsymbol{v}_i$ have the same norm, the crucial factor for approximation quality is reduced to the attention weights $\alpha_i$. In this context, we also call $\alpha_i$ the *contribution* of key-value pair $i$.

Let $K$ be a positive integer. The general idea of all methods discussed in this work is to keep $K$ pairs $(\boldsymbol{k}_i, \boldsymbol{v}_i)$ whose contribution $\alpha_i$ is the highest, and ignore other low-contribution pairs. The goal is to find the best mechanism to select such $K$ pairs. Here we discuss three variants: Top-$K$ activation (Sec. 3.1), Product-Key Memories (PKMs, Sec. 3.2), and Mixture of Experts (MoEs, Sec. 3.3).

### 3.1 Top-$K$ Activation Function

The most straightforward implementation of the approximation described above is the top-$K$ activation function:

$$\mathcal{E}_{\boldsymbol{x}} = \arg\text{topk}(\boldsymbol{u}, K) \subset \{1, ..., d_{\text{ff}}\} \quad (6)$$

$$\hat{\boldsymbol{y}} = \sum_{i \in \mathcal{E}_{\boldsymbol{x}}} \alpha_i \boldsymbol{v}_i \quad (7)$$

Unfortunately this only saves less than half of the entire computation: while this allows us to reduce computation of Eq. 2, no computation can be saved in Eq. 1 because full computation of $\boldsymbol{u} =$

ReLU $(\boldsymbol{W}_1 \boldsymbol{x})$ is required for Eq. 6. Going beyond this requires to also introduce some approximation to Eq. 6 as in PKMs (Sec. 3.2) and MoEs (Sec. 3.3).

### 3.2 Product-Key Memories (PKMs)

Product-Key memories (Lample et al., 2019) consist of replacing $\boldsymbol{W}_1 \in \mathbb{R}^{d_{\text{ff}} \times d_{\text{model}}}$ in Eq. 1 by two matrices $\boldsymbol{W}_a, \boldsymbol{W}_b \in \mathbb{R}^{\sqrt{d_{\text{ff}}} \times \frac{d_{\text{model}}}{2}}$. It slices the input vector $\boldsymbol{x} \in \mathbb{R}^{d_{\text{model}}}$ into two halves, $\boldsymbol{x}_a, \boldsymbol{x}_b \in \mathbb{R}^{\frac{d_{\text{model}}}{2}}$, so that $\boldsymbol{x} = \boldsymbol{x}_a | \boldsymbol{x}_b$, where $|$ denotes concatenation. The matrix multiplication is then performed on these smaller vectors: $\boldsymbol{u}_a = \boldsymbol{W}_a \boldsymbol{x}_a$ and $\boldsymbol{u}_b = \boldsymbol{W}_b \boldsymbol{x}_b$. Then $\boldsymbol{u} \in \mathbb{R}^{d_{\text{ff}}}$ is calculated by combining the elements of $\boldsymbol{u}_a \in \mathbb{R}^{\sqrt{d_{\text{ff}}}}$ and $\boldsymbol{u}_b \in \mathbb{R}^{\sqrt{d_{\text{ff}}}}$ in all possible ways (i.e., Cartesian products), similarly to the outer product, but using addition instead of multiplication, i.e., for all $i \in \{1, ..., d_{\text{ff}}\}$,

$$\boldsymbol{u}[i] = \boldsymbol{u}_b[\lfloor i/\sqrt{d_{\text{ff}}} \rfloor] + \boldsymbol{u}_a[i \bmod \sqrt{d_{\text{ff}}}] \quad (8)$$

In addition to applying Top-$K$ at the output as in Sec 3.1, here Top-$K$ can also be used to accelerate the operation above. By applying Top-$K$ to $\boldsymbol{u}_a$ and $\boldsymbol{u}_b$ before combining them to compute $\boldsymbol{u}$, only the $K^2 << d_{\text{ff}}$ components of $\boldsymbol{u}[i]$ have to be calculated, and they are guaranteed to contain the $K$ biggest components of the full $\boldsymbol{u}$.

In the original formulation (Lample et al., 2019), PKMs use a softmax activation function, taking inspiration from self-attention (Vaswani et al., 2017). Instead, we'll show how a non-competing activation function, such as ReLU is a better choice (see Sec. 6.2).

### 3.3 Mixture of Experts (MoE)

Let $N_E, G$ denote positive integers. MoEs partition $d_{\text{ff}}$ pairs of $(\boldsymbol{k}_i, \boldsymbol{v}_i)$ (see their definition in Sec. 2) into $N_E$ groups of size $G$ each, such that $G \cdot N_E = d_{\text{ff}}$. This means that the weight matrices $\boldsymbol{W}_1 \in \mathbb{R}^{d_{\text{ff}} \times d_{\text{model}}}$ and $\boldsymbol{W}_2 \in \mathbb{R}^{d_{\text{model}} \times d_{\text{ff}}}$ (Eqs. 1-2) are partitioned into matrices $\boldsymbol{W}_1^e \in \mathbb{R}^{\frac{d_{\text{ff}}}{N_E} \times d_{\text{model}}}$ and $\boldsymbol{W}_2^e \in \mathbb{R}^{d_{\text{model}} \times \frac{d_{\text{ff}}}{N_E}}$ for $e \in \{1, ..., N_E\}$,

$$\boldsymbol{W}_1^e = \begin{bmatrix} — & \boldsymbol{k}_{eG+1}^{\mathsf{T}} & — \\ — & \boldsymbol{k}_{eG+2}^{\mathsf{T}} & — \\ & \vdots & \\ — & \boldsymbol{k}_{(e+1)G}^{\mathsf{T}} & — \end{bmatrix} \quad (9)$$

$$\boldsymbol{W}_2^e = \begin{bmatrix} | & | & & | \\ \boldsymbol{v}_{eG+1} & \boldsymbol{v}_{eG+2} & \cdots & \boldsymbol{v}_{(e+1)G} \\ | & | & & | \end{bmatrix}$$

$$(10)$$

The output is computed as:

$$\hat{\boldsymbol{y}} = \sum_{e \in \mathcal{E}_{\boldsymbol{x}}} \boldsymbol{W}_2^e \boldsymbol{s}[e] \, \mathrm{ReLU}(\boldsymbol{W}_1^e \boldsymbol{x}) \qquad (11)$$

where $\boldsymbol{s}[e] \in \mathbb{R}$ is the $e$-th element of vector $\boldsymbol{s} \in \mathbb{R}^{N_E}$ computed by an expert scoring function $\mathrm{sel} : \mathbb{R}^{d_{\mathrm{model}}} \to \mathbb{R}^{N_E}$ (typically $\boldsymbol{s} = \mathrm{sel}(\boldsymbol{x}) = \mathrm{softmax}(\boldsymbol{W}_3 \boldsymbol{x})$ with $\boldsymbol{W}_3 \in \mathbb{R}^{N_E \times d_{\mathrm{model}}}$), and $\mathcal{E}_{\boldsymbol{x}}$ denotes a subset of indices $\{1, ..., N_E\}$ resulting from the Top-$K$ operation on $\boldsymbol{s}$, i.e., $\mathcal{E}_{\boldsymbol{x}} = \arg \mathrm{topk}(\boldsymbol{s}, K)$. Note that in some variants, additional *re-normalization* is applied *after* Top-$K$, so that $\sum_{e \in \mathcal{E}_{\boldsymbol{x}}} \boldsymbol{s}[e] = 1, \boldsymbol{s}[e] \geq 0$; we define such an operation as $\mathrm{norm\,topk}$, see its exact definition in App. A.1 [4]. The efficiency of MoEs comes from the fact that $N_E \ll d_{\mathrm{ff}}$, thus calculating $\boldsymbol{s}$ is cheap. Furthermore, $G$ and $K$ are chosen so that $G * K \ll d_{\mathrm{ff}}$, so the calculation performed by experts is less expensive than the dense MLP.

Given the notation above, it is straightforward to see that MoEs can also be viewed as approximating 2-layer MLPs with a trainable component (i.e., the selection function $\mathrm{sel}$ to produce $\boldsymbol{s}$). Similarly to Eqs. 5 and 7, Eq. 11 can be expressed as:

$$\hat{\boldsymbol{y}} = \sum_{e \in \mathcal{E}_{\boldsymbol{x}}} \sum_{i=1}^{G} \alpha_{eG+i} \boldsymbol{s}[e] \boldsymbol{v}_{eG+i} \qquad (12)$$

where, compared to Eqs. 5 and 7, the "contribution scores" of key-value pair $i$ (defined in Sec. 3/Preliminaries) have an additional factor $\boldsymbol{s}[e]$ of an expert group $e$ to which the key-value pair belongs.

The key challenge of MoEs is to learn an expert selection mechanism/function $\mathrm{sel}$ above that assigns high scores to only a few experts (so that we can ignore others without sacrificing performance), while avoiding a well-known issue, called expert *collapsing*, where only a few experts are used and the rest are never selected. To avoid this, some regularization is typically applied to the selection score $\mathrm{sel}(\boldsymbol{x})$, encouraging more uniform routing of experts across the whole batch of tokens. We provide a comprehensive review of MoE variants and their details in Sec. 4 and our improved version in Sec. 5.

## 4 Existing MoE variants

Several variations of MoEs have been proposed with many different details. Here we briefly review the most popular and representative ones (e.g., we

---

[4]In the case of the $\mathrm{softmax}(\cdot)$ activation function, this is equivalent to applying Top-$K$ to the logits *before* softmax.

do not cover those that make use of reinforcement learning for expert routing) before describing our improved version in Sec. 5. We'll review their *expert selection function* and *regularization method*, and highlight their key characteristics.

**Sparsely Gated Mixtures of Experts.** Shazeer et al. (2017) have revisited MoEs (Jacobs et al., 1991; Ivakhnenko and Lapa, 1965) with the Top-$K$ operation, allowing a reduction in its resource demands. Their method is basically the one described in Sec. 3.3 (with re-normalization after Top-$K$) except that they use a noisy gating function:

$$\mathrm{sel}(\boldsymbol{x}) = \mathrm{softmax}($$
$$\boldsymbol{W}_3 \boldsymbol{x} + \mathcal{N}(0,1) \cdot \mathrm{softplus}(\boldsymbol{W}_4 \boldsymbol{x})) \quad (13)$$

where $\boldsymbol{W}_4 \in \mathbb{R}^{N_E \times d_{\mathrm{model}}}$, the Gaussian noise term $\mathcal{N}(0,1)$ is element-wise and independent for each channel, and $\mathrm{softplus}(x) = \log(1 + e^x)$. They use the following auxiliary regularization term for load balancing,

$$L = \mathrm{CV} \left( \sum_{\boldsymbol{x} \in \mathcal{B}} \mathrm{norm\,topk}(\mathrm{sel}(\boldsymbol{x})) \right) \qquad (14)$$

where $\mathrm{CV}(x) = \frac{\mu_x}{\sigma_x}$ is the coefficient of variation and $\mathcal{B}$ is the set of all tokens in the batch.

**Key characteristics:** The scores are normalized after the top-$K$ operation (with $K > 1$), which is equivalent to applying top-$K$ *before* the softmax.

**Switch Transformer.** Fedus et al. (2022) integrate the MoE above into the Transformer to obtain their Switch Transformer. In terms of MoE details, one of Fedus et al. (2022)'s key claims is that top-1 routing is enough. Their selection function is simply: $\mathrm{sel}(\boldsymbol{x}) = \mathrm{softmax}(\boldsymbol{W}_3 \boldsymbol{x})$, but they propose a hard load-balancing between experts that run on different hardware accelerators: At most $\mu \frac{|\mathcal{B}|}{N_E}$ tokens are allowed to be routed to an expert, where $\mu \in \mathbb{R}_{>0}$ is the capacity factor (typically between 1 and 1.5), defining how many times more tokens can be processed by one expert compared to the ideal case of uniform routing. Each expert is forbidden to process more than this number of tokens. For regularization, the fraction of the tokens $\boldsymbol{f} \in \mathbb{R}^{N_E}$ processed by each expert, and the average selection probability $\boldsymbol{p} \in \mathbb{R}^{N_E}$ for each expert are calculated

($K = 1$; top-1 is used) as:

$$\boldsymbol{f}_i = \frac{1}{|\mathcal{B}|} \sum_{\boldsymbol{x} \in \mathcal{B}} \mathbb{1}\{i \in \arg \operatorname{topk}(\operatorname{sel}(\boldsymbol{x}), K)\} \quad (15)$$

$$\boldsymbol{p} = \frac{1}{|\mathcal{B}|} \sum_{\boldsymbol{x} \in \mathcal{B}} \operatorname{sel}(\boldsymbol{x}) \quad (16)$$

$$L = N_E \boldsymbol{f} \cdot \boldsymbol{p} \quad (17)$$

where $\mathbb{1}$ denotes the indicator function (which is equal to 1 if the argument is true, and 0 otherwise), and $\cdot$ denotes dot product. Intuitively, this serves as an adaptive regularization that penalizes experts that are used often with high "weights." In addition, they use dropout with a high drop rate (40%) in the experts (but only 10% in the normal layers).

Furthermore, Fedus et al. (2022) also propose to initialize the experts with $\sqrt{\frac{0.1}{G}}$. As we'll see in Sec. 5, we use a modified version of this scheme.

Note that applying Top-$K$ after softmax encourages collapsing: if the score of the selected expert is increased, the scores of all other experts are automatically decreased. This is not the case for Shazeer et al. (2017): In their method, only the selected experts compete with each other, so if their presence is beneficial, their score can be increased.

**Key characteristics:** Note that Top-1 is applied *after* the softmax without re-normalization.

**BASE layers and S-BASE.** Inspired by the routing strategy and the hard capacity factor of the Switch Transformer, Lewis et al. (2021) propose BASE layers. They use top-1 routing and a sigmoid activation $\sigma$ in the selection function:

$$\operatorname{sel}(\boldsymbol{x}) = \sigma(\boldsymbol{W}_3 \boldsymbol{x}) \quad (18)$$

Now instead of using $\arg \operatorname{topk}$, they solve the following linear assignment problem to find the index $e_{\boldsymbol{x}} \in \{1, ..., N_E\}$ of the expert to which each input $\boldsymbol{x} \in \mathcal{B}$ is routed,

$$\operatorname*{maximize}_{e_{\boldsymbol{x}} \in \{1,...,N_E\}, \boldsymbol{x} \in \mathcal{B}} \sum_{\boldsymbol{x} \in \mathcal{B}} \operatorname{sel}(\boldsymbol{x})[e_{\boldsymbol{x}}] \quad (19)$$

$$\text{s.t. } \forall i \in \{1, ..., N_E\}, \sum_{\boldsymbol{x} \in \mathcal{B}} \mathbb{1}\{e_{\boldsymbol{x}} == i\} = \frac{|\mathcal{B}|}{N_E}$$

This guarantees uniform assignment of experts, which is efficient for multi-accelerator training. The output is computed using Eq. 11 with $\mathcal{E}_{\boldsymbol{x}} = \{e_{\boldsymbol{x}}\}$ (a set with a single element; "top-1"). However, at inference time, no such balancing is possible because not all tokens of the sequence are available at each step; $\mathcal{E}_{\boldsymbol{x}} = \{\arg \max (\operatorname{sel}(\boldsymbol{x}))\}$ is used instead. Lewis et al. (2021) show that, while during training, the routing is enforced to be completely uniform, during the test time, the distribution looks exponential (in fact, this is similar to the Switch Transformer but more balanced for BASE).

The algorithm for solving the linear assignment problem (Eq. 19) is difficult to implement efficiently on modern accelerators. Clark et al. (2022) have proposed to use the Sinkhorn algorithm (Sinkhorn, 1964; Sinkhorn and Knopp, 1967) instead (resulting in a model called Sinkhorn-BASE or S-BASE), to approximate the solution to this problem (note that similar routing is independently discussed by Kool et al. (2021)). They report that this works well, while being simpler to implement. Thus, our reimplementation of BASE is S-BASE using the Sinkhorn algorithm.

**Key characteristics:** During training, Sinkhorn iterations are used on scores to obtain a balanced assignment. The sigmoid activation is always applied to compute the weighting score.

**Overall**, all load-balancing methods above are rather complex. We propose simpler but effective approach for MoEs in Sec. 5.

## 5 Improving Mixture of Experts

Here we present our improved MoE variant, which we call $\sigma$-MoE. We conduct thorough ablation studies on our design choices in Sec. 6.

$\sigma$-**MoE Expert Selection Function.** Our MoE make use of the top-$K$ operation (unlike BASE). The activation we use on the selection function is sigmoid (as in Eq. 18 of BASE) instead of softmax used in Switch Transformer and Sparsely Gated Mixtures of Experts. This choice is motivated by the view of MoEs as approximate 2-layer MLPs (Sec. 3). In fact, softmax introduces competition between experts. No such competition between channels is used in the regular 2-layer MLP (i.e., there is no constraint on $\alpha_i$ in Eq. 5). This suggests that, in principle, no competition is needed between terms in the sum of Eq. 12 in the MoE either, to induce sparsity. It is also well known to practitioners that softmax as regular activation negatively affects the trainability of standard MLPs. Softmax combined with top-$K$ can also encourage expert collapsing: when the selection score of one expert increases, the score of the others automatically decreases. For all these reasons, we opt for

sigmoid instead of softmax; we experimentally confirm that this is indeed a good choice.

Additionally, looking at MoEs in this framework gives us hints on combining them with Top-$K$ activation (Sec. 3.1) for further acceleration. We can calculate $\boldsymbol{u}^e = \boldsymbol{s}[e]\,\mathrm{ReLU}(\boldsymbol{W}_1^e x)$ (Eq. 11) for the selected experts and perform an additional Top-$K$ to keep the highest units among them and set the rest to zero. We leave this for future work.

**$\sigma$-MoE Initialization.** Another design choice guided by the MLP-approximation view of MoEs (Sec. 3) is the initialization scheme for experts. Typically, experts are assumed to be independent, and the standard deviation of the initialization (Glorot and Bengio, 2010; He et al., 2015) of $\boldsymbol{W}_2^e$ is calculated based on $G$ instead of $d_{\mathrm{ff}}$. Our experiments in Sec. 6.3 show that this is sub-optimal.

In contrast, we initialize all weight matrices identically to the pre-layernorm dense baselines, not taking in account the smaller size of the individual experts, i.e., $\boldsymbol{W}_1^e \sim \mathcal{N}(0, \sqrt{\frac{2}{d_{\mathrm{model}} \cdot n_{\mathrm{layers}}}})$ and $\boldsymbol{W}_2^e \sim \mathcal{N}(0, \sqrt{\frac{2}{d_{\mathrm{ff}} \cdot n_{\mathrm{layers}}}})$ where $n_{\mathrm{layers}}$ denotes the number of layers, using $d_{\mathrm{model}}$ and $d_{\mathrm{ff}}$ instead of $G$.

We also take special care when initializing $\boldsymbol{W}_3$ of the selection function. We initialize it to a normal distribution with the same standard deviation as $\boldsymbol{W}_1^e$, but we also ensure that the rows of $\boldsymbol{W}_3$ have the same norm[5]. This can be easily achieved in practice by initializing the weights to $\boldsymbol{W}_3' \sim \mathcal{N}(0, 1)$, rescaling its rows to norm 1, and then rescaling the whole matrix again to have the desired standard deviation. Note that each scalar score in $\boldsymbol{s}$ is the dot product of a row of $\boldsymbol{W}_3$ and $\boldsymbol{x}$. This initialization method ensures that only the angle between $\boldsymbol{x}$ and the rows of $\boldsymbol{W}_3$ initially affects the score $\boldsymbol{s}$, rather than an additional random factor resulting from initialization.

**$\sigma$-MoE Regularization.** As already noted in Sec. 4, existing regularization methods for load-balancing are complex (e.g., Switch Transformers need to deal separately with the actual selection distribution and the scores, Sparsely Gated Mixture of Experts needs noise in the selection function). In contrast, we propose to simply maximize the entropy of the selection distribution $\boldsymbol{p} \in \mathbb{R}^{N_E}$ calculated across the entire batch. Intuitively, this is a

simple way to encourage equal expert usage within the batch and prevent unnecessary overconfidence in selecting individual experts. Let $\mathcal{B}$ be the set of all tokens in the batch (counting through both the batch and time dimensions). We introduce the following regularization term $L$:

$$\boldsymbol{p} = \frac{1}{|\mathcal{B}|} \sum_{\boldsymbol{x} \in \mathcal{B}} \mathrm{softmax}(\boldsymbol{W}_3 \boldsymbol{x}) \qquad (20)$$

$$L = \sum_{e=1}^{N_E} \boldsymbol{p}[e] \log \boldsymbol{p}[e] \qquad (21)$$

Furthermore, we propose to randomly drop complete experts, during training; we refer to this as *expert dropout*. Unlike the standard dropout on the activation level, we do not apply rescaling, i.e.,

$$\mathrm{sel}(\boldsymbol{x}) = \begin{cases} \sigma(\boldsymbol{W}_s \boldsymbol{x}) \odot \boldsymbol{m} & \text{if training} \\ \sigma(\boldsymbol{W}_s \boldsymbol{x}) & \text{otherwise} \end{cases} \qquad (22)$$

where $\boldsymbol{m} \in \{0, 1\}^{N_E}$, $\boldsymbol{m} \sim \mathrm{Bernoulli}(1 - \delta)$, where $\delta$ is the dropout rate, and $\odot$ is the element-wise product. This prevents the dropped experts from being selected while not affecting the other ones. Intuitively, when an expert dropout removes a popular expert, it forces the less popular ones to take over. Thus, the chance of them being trained and improved increases. We experimentally show that our regularization method (Eq. 21) and expert dropout (Eq. 22) are both effective despite their simplicity.

## 6 Experiments

Our experimental setup is based on Dai et al. (2019)'s Transformer XL with some modifications: we use pre-layer norm and reduce the number of training steps to 100k to reduce the computational budget. Also, to match the parameter counts between the baseline and MoEs, we slightly modify the hyperparameters of the baselines (Dai et al., 2019). In fact, our MoE CUDA kernel can only work with dimensions divisible by 4. We round the original sizes up to the next suitable number, e.g., we change $d_{\mathrm{model}}$ of our 47M-parameter WikiText-103 model from the original 410 to 412. Furthermore, since MoEs require extra parameters for the expert selection function, we compensate for these by increasing the $d_{\mathrm{ff}}$ of the baseline model to match the number of parameters. Our modified baseline model on Enwik8 still has 41M parameters and performs similarly to the original Transformer XL

---

[5]Having rows with different norms would discourage the use of experts corresponding to rows with small norms, as their selection score would be low even if the angle of the selector (row of $\boldsymbol{W}_3$) fully aligns with $\boldsymbol{x}$.

(see Tab. 1). For WikiText-103, we use subword units (Sennrich et al., 2016) using SentencePiece tokenizer (Kudo and Richardson, 2018) instead of the word-level vocabulary, to avoid extra tricks required to reduce the parameter count and compute requirement resulting from the huge vocabulary size. On WikiText-103, we consider two different model sizes: a 47M-parameter one (denoted by "WT-S" for "small"), and a 262M-parameter one ("WT-B" for "big"). We refer to Enwik8 as "E8" in certain tables. For more details, see Appendix B.

For all the methods considered, we use them in *every* MLP block of the model, which is not a common practice in the literature. Typically, MoE (or other approximation methods) is used only once every $n^{th}$ layer or even only in one layer. This is not satisfactory since our goal is to find a generally applicable method that can accelerate all layers across the whole model. Moreover, this amplifies the difference between different methods, helping better illustrate effects of each of the design choices.

## 6.1 Top-$K$

We first evaluate the Top-$K$ method (Sec. 3.1). This standalone evaluation is important as Top-$K$ is the basis of both the PKM and the MoE approximations. Tab. 1 shows the results. We observe that not only Top-$K$ in the MLP blocks preserves the performance of Transformers, it even improves performance. We hypothesize that these improvements are due to the reduction in feature interference as described by Elhage et al. (2022). However, we obviously can not arbitrarily reduce $K$; there should be a trade-off between the denoising effect and the capacity of the network. Here, the optimal value we find is $K = 128$ or $K = 512$.

Table 1: Effects of the top-k activation function on the perplexity (WikiText-103) and bits/character (Enwik8).

| Dataset | #params | $d_{ff}$ | $K$ | bpc/perplexity |
|---------|---------|----------|-----|----------------|
| Enwik8 | 41M | 2053 | - | 1.08 |
| | 41M | 2053 | 128 | **1.07** |
| | 41M | 2053 | 256 | 1.08 |
| | 41M | 2053 | 512 | 1.08 |
| WikiText 103 | 47M | 2053 | - | 11.81 |
| | 47M | 2053 | 64 | 11.86 |
| | 47M | 2053 | 128 | 11.74 |
| | 47M | 2053 | 256 | 11.74 |
| | 47M | 2053 | 512 | **11.68** |
| WikiText 103 | 262M | 4110 | - | 9.46 |
| | 262M | 4110 | 128 | **9.26** |
| | 262M | 4110 | 256 | 9.34 |
| | 262M | 4110 | 512 | 9.36 |

## 6.2 Product-Key Memory (PKM)

Our view of Sec. 3 suggests using a non-competitive activation such as ReLU instead of the softmax used in the original PKM (Lample et al., 2019). Our experiments confirm the benefits of this choice (Tab. 2): the performance of the ReLU variants is much closer to the dense baseline (see also related findings in Shen et al. (2023)). But even the best PKM models underperform the dense baselines, indicating the fundamental limitation of PKMs. Note that, as stated above, we conduct a careful comparison between the approximation method (here, PKM) and the dense baseline using the same number of parameters. For more results and details on PKM, we refer to App. A.3.

Table 2: Performance of the parameter-matched PKM models. We provide more results in Appendix/Tab. 6.

| Variant | Nonlin | WT-S | WT-B | E8 |
|---------|--------|------|------|-----|
| Dense Baseline | ReLU | 11.81 | 9.46 | 1.08 |
| PKM | Softmax | 13.96 | 11.10 | 1.16 |
| | ReLU | 12.77 | 9.98 | 1.11 |

## 6.3 Mixture of Experts (MoE)

Here we evaluate our $\sigma$-MoE models (Sec. 5) on Enwik8 and WikiText-103 as well as two additional datasets, C4 (Raffel et al., 2020) and the newly proposed peS2o (Soldaini and Lo, 2023). Given the large sizes of C4 and peS2o, we cannot afford to train for a full epoch; we train for 100k steps with the same hyperparameters as for WikiText-103.

**Main results.** Tab. 3 shows the main results. Our $\sigma$-MoE models match the performance of their parameter-equal dense baselines, while achieving significant memory and compute reduction. These models use $K = 4$ for $N_E = 16$ or $N_E = 32$, which is a "moderate" level of sparsity but already offering significant compute reduction as shown in the column "% FLOPs"; concrete compute and memory reduction is further shown in Fig. 2 (see Appendix A.5 for details). Naturally, there is a limit on the minimum sparsity level to preserve good performance of MoEs, which is determined by several factors. First, we empirically find that experts with a group size of $G < 128$ generally degrades performance. Second, our benchmarks with the Top-$K$ operation (Tab. 1) and our ablations (Tab. 10 in the Appendix) show that the minimum number of simultaneously active

Table 3: Performance of parameter-batched $\sigma$-MoEs on perplexity (WikiText-103, C4 and peS2o) and bits/character (Enwik8). Ours matches or surpasses the performance of the dense baselines across all datasets.

| Dataset | Model | #params | % FLOPs | bpc/ppl |
|---|---|---|---|---|
| Enwik8 | Dense | 41M | 100.0% | **1.08** |
| | $\sigma$-MoE | 41M | 25.0% | **1.08** |
| WikiText-103 | Dense | 47M | 100.0% | 11.81 |
| | $\sigma$-MoE | 47M | 25.0% | **11.71** |
| WikiText-103 | Dense | 262M | 100.0% | 9.46 |
| | $\sigma$-MoE | 262M | 12.5% | **9.44** |
| C4 | Dense | 47M | 100.0% | 23.76 |
| | $\sigma$-MoE | 47M | 25.0% | **23.25** |
| C4 | Dense | 262M | 100.0% | 17.79 |
| | $\sigma$-MoE | 262M | 12.5% | **17.46** |
| peS2o | Dense | 47M | 100.0% | 14.34 |
| | $\sigma$-MoE | 47M | 25.0% | **14.12** |
| peS2o | Dense | 262M | 100.0% | **10.91** |
| | $\sigma$-MoE | 262M | 12.5% | 10.91 |

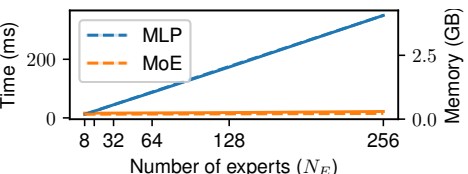

Figure 2: Execution time and memory usage of a forward-backward pass of a single MLP and MoE layer. $|\mathcal{B}| = 32768$, corresponding to a batch size 64 and sequence length 512, $d_{\text{model}} = 512$, $K = 4$, and $d_{\text{ff}} = G \cdot N_E$. Full/dashed lines show the execution time/memory consumption, respectively. As they are both linear with similar slopes, they are almost indistinguishable. Even our sub-optimal CUDA kernel is faster starting from 16 experts. Measured on an RTX 3090 with PyTorch 2.01 and CUDA 11.

channels $G \cdot K$ need to be above a certain critical threshold (usually around 256-512). Finally, we match the number of parameters of the baseline model; this is the last constraint. Under these constraints, we find that the performance of the dense baselines can be matched using 25% of the required FLOPs and memory for activations for our small models, and 12.5% sparsity for the big one (note that FLOPs here do not take into account the linear projection used to select the experts, which is negligible within the range of $N_E$ used here).

**Increasing $N_E$ and Impact of Sparsity.** The results above demonstrate that our $\sigma$-MoEs can be configured to match the desired performance with fewer resources. Here we conduct an extra experiment where we naively increase $N_E$ (while keeping $K = 4$) from 16 to 128. This increases the number of parameters to 238M, while keeping the speed and memory requirements comparable to the original model (column "WT-S*" in Tab. 4). This model achieves a test perplexity of 10.37, which is worse than 9.46 of the 262M dense model (see Tab. 1). Indeed, even when the parameter count is matched, there are other bottlenecks that are crucial, e.g., here $d_{\text{model}}$ is much smaller (412 vs 1024). We construct another dense baseline by setting every hyperparameter like in the 47M model, except $d_{\text{ff}}$, which is set to 16480 to match the number of parameters of the $N_E = 128$ MoE. This baseline achieves a perplexity of 10.03: thus, the gap between the scaled-up MoE and its dense counterpart

still remains significant (10.37 vs 10.03), unlike with the MoE with moderate sparsity. This indicates the importance of controlling MoE sparsity to preserve its performance against the dense baseline.

**Comparison to Existing MoEs.** We also compare our $\sigma$-MoE to other MoE variants (Sec. 4), namely Switch Transformer (Fedus et al., 2022), S-BASE (Clark et al., 2022)[6] and the basic softmax variant. Tab. 4 shows the results for multiple variants on WikiText-103 and Enwik8. Additionally, we compare $\sigma$-MoE to the most important baselines on C4 and peS2o in Tab. 5. As Switch Transformer and S-BASE select only one single expert ($K = 1$), we increase the expert size by a factor of 4 (instead of $G = 128$ in our models, we use $G = 512$), and we decrease $N_E$ by the same factor for fair comparison in terms of the parameter count. Neither of them uses our proposed expert dropout. For Switch Transformer, we test a variant with standard dropout in the experts (see App. B for details), and a version without. We also extend S-BASE to $K = 4$, which is similar to ours, except for the balancing method. Even considering all these cases, our $\sigma$-MoE outperforms Switch Transformer and S-BASE. Note that in terms of FLOPs and memory usage, all MoE variants are equivalent given the same hyperparameters ($G$, $d_{\text{model}}$, and $K$).

**Ablation Studies.** Finally we conduct ablation studies of individual design choices (Sec. 5). Tab. 4 shows the results. Standard dropout instead of expert dropout leads to performance degradation for most of the cases, except the model with $N_E = 128$ experts. The softmax-based selection functions

---

[6]Unlike the original ones, our implementation does not enforce capacity factor-based hard balancing.

Table 4: Ablation studies. WT-S* is obtained by naively scaling $N_E$ in WT-S. More details in Sec. 6.3 & Tab. 10.

| Dataset | WT-S | WT-S* | WT-B | E8 |
|---|---|---|---|---|
| # params. (in M) | 47 | 238 | 262 | 41 |
| Switch Transformer | 12.27 | 11.24 | 9.68 | **1.08** |
| no dropout | 11.88 | 11.10 | 9.77 | 1.10 |
| S-BASE ($K$=4, $G$=128) | 13.01 | 10.96 | 10.50 | 1.17 |
| $K=1, G=512$ | 12.32 | 11.31 | 9.77 | 1.32 |
| $\sigma$-MoE ($K$=4, $G$=128) | **11.59** | 10.37 | **9.44** | **1.08** |
| standard dropout | 12.01 | **10.27** | 9.53 | **1.08** |
| softmax (renorm.) | 11.89 | 11.27 | 9.58 | 1.09 |
| softmax (no renorm.) | 12.05 | 10.54 | 9.62 | 1.09 |
| standard init | 11.80 | 10.59 | 9.67 | **1.08** |
| no regularization | 11.83 | 10.41 | 9.51 | **1.08** |
| $K=8, G=64$ | 11.63 | 10.30 | 9.58 | **1.08** |
| $K=2, G=256$ | 11.84 | 10.44 | 9.56 | 1.09 |
| $K=1, G=512$ | 11.90 | 10.83 | 9.58 | 1.09 |

Table 5: Perplexity of $\sigma$-MoE compared to parameter-matched baselines on C4 and peS2o datasets.

| Dataset | | | C4 | C4 | peS2o | peS2o |
|---|---|---|---|---|---|---|
| $d_{model}$ | | | 412 | 1024 | 412 | 1024 |
| # params | | | 47M | 262M | 47M | 262M |
| | G | K | | | | |
| Dense | 128 | 1 | 23.76 | 17.79 | 14.34 | **10.91** |
| $\sigma$-MoE | 128 | 4 | **23.25** | **17.46** | **14.12** | **10.91** |
| Switch | 512 | 1 | 24.47 | 18.29 | 14.74 | 11.56 |
| S-BASE | 128 | 4 | 35.48 | 18.53 | 16.61 | 11.72 |

(with and without re-re-normalization) consistently perform worse than our sigmoid one. The same is true for the standard initialization ; ours is better. Interestingly, removing all regularization methods degrades performance but does not entail catastrophic collapse even with $N_E = 128$. We also examine the best $(G, K)$ combinations, given a constant number $(G \cdot K)$ of active pairs $\boldsymbol{k}_i, \boldsymbol{v}_i$; we find a high $K = 4$ works best within this range. Further analysis of our $\sigma$-MoE can be found in App. A.4.

**Analyzing expert utilization.** A typical failure mode of MoEs is expert collapse, where only a few experts are used while others are completely ignored or underused. Here we conduct an analysis to evaluate whether various models including ours are affected by this issue. For each layer, we compute the proportion of the expert selection weights assigned to each expert $(\text{sel}(\boldsymbol{x}))$ on the entire validation set of WikiText-103. We use WT-S* models from Tab. 4 with 128 experts. A representative layer is shown in Fig. 3. Models with poor performance (see Tab. 4), i.e., Switch Transformer (*red*) and a "bad" variant of $\sigma$-MoE with a softmax and renormalization "softmax (renorm.)" (*green*),

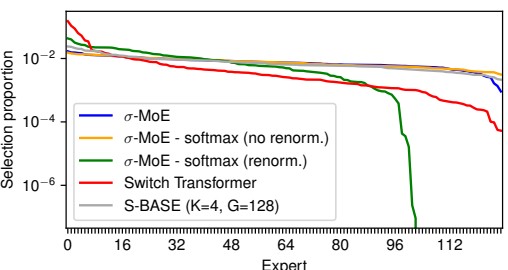

Figure 3: The total proportion of selection weights assigned to a given expert (x-axis; sorted by their popularity) on the validation set of Wikitext-103 using the WT-S* models from Tab. 4. This is for one representative layer ("Layer 5"; similar plots for other layers are shown in Fig. 7 in the appendix). The models with poor performance (Tab. 4), i.e., Switch Transformer (*red*) and $\sigma$-MoE with a softmax and renormalization "softmax (renom.)" (*green*) can be easily identified. In contrast, the fine performance differences between the rest of the models do not seem to be due to expert collapse.

can be easily identified: they severely suffer from the expert collapse problem. The statistics are rather similar for all other models; the fine performance differences among these models do not seem to be due to expert collapse. Remarkably, our entropy-regularized models with expert dropout, especially $\sigma$-MoE, are capable of matching the expert usage balancing of S-BASE without using the Sinkhorn activation function. Note that in general, we do not consider uniform expert activation to be optimal: we expect expert specialization, and thus the frequency of their usage should depend on the occurrence of the task they are performing.

## 7 Conclusion

Our novel view unifies methods that approximate 2-layer MLPs, such as Top-$K$, Mixture of Experts (MoE) and product-key memory (PKM) methods. While Top-$K$ by itself provides limited performance improvements and speedups, further speedup requires PKM or MoE. A non-competitive activation function inspired by our unified view improves both PKM and MoE. Further novel enhancements of MoEs yield our $\sigma$-MoE which outperforms existing MoEs. Importantly, our $\sigma$-MoE with moderate sparsity matches the performance of parameter-equal dense baselines while being much more resource-efficient. Our new insights improve the training of language models with limited hardware resources, making language modeling research more accessible.

## Limitations

Our experiments show that if we naively increase the number of experts, the performance gap between MoE models and their dense counterparts increases. This indicates the need for careful control of sparsity and hyper-parameters, which remains a challenge for MoEs.

Our CUDA kernel is sub-optimal and I/O limited. However, even in its current form, it already yields significant performance boosts and memory reduction. We expect that an expert CUDA programmer could improve the speed of our kernel by at least a factor of 2.

We do not consider load balancing between hardware accelerators as is done in Switch Transformers and S-BASE. Our goal is to make a larger model fit a single accelerator, or multiple accelerators in the standard data-parallel training. Our preliminary experiments suggest that such balancing entails a performance hit.

We could not reproduce the 277M Enwik8 model of Dai et al. (2019), because we could not fit the beaseline model on any of our machines. We tried to use rotary positional encodings with PyTorch 2.0's memory-efficient attention to reduce it's memory consumption; however, this resulted in a significant performance degradation (even for the smaller models).

Our study focuses on end-to-end trainable MoEs. Other MoE methods (Irie et al., 2018; Li et al., 2022) that pre-train LMs on disjoint data, to recombine them later into a single model, are out-of-scope.

Our study only considers standard Transformers; however, similar acceleration methods are of utmost importance for shared-layer Transformers, such as Universal Transformers (Dehghani et al., 2019) and NDRs (Csordás et al., 2022). In fact, layer sharing dramatically reduces the number of parameters. Compensating for this by naively increasing $d_{\mathrm{model}}$ or $d_{\mathrm{ff}}$ results in prohibitively high memory overhead and slow execution. In contrast, MoEs allow increasing the number of parameters without such dramatic drawbacks. We leave shared-layer MoEs for future work.

## Acknowledgements

This research was partially funded by ERC Advanced grant no: 742870, project AlgoRNN, and by Swiss National Science Foundation grant no: 200021_192356, project NEUSYM. We are thankful for hardware donations from NVIDIA and IBM. The resources used for this work were partially provided by Swiss National Supercomputing Centre (CSCS) project s1154, s1205 and d123. Finally, we would like to thank Dániel Berényi for his support with the CUDA kernel development.

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

## A Further details and analyses

### A.1 Definition of normalised Top-$K$

Using the setting of Sec. 3.3, we define the *normalized top-$K$* operation as follows:

$$\mathcal{E}_{\boldsymbol{x}} = \arg \operatorname{topk}(\boldsymbol{s}, K) \tag{23}$$

$$\operatorname{topk}(\boldsymbol{s})[i] = \begin{cases} \boldsymbol{s}[i] & \text{if } i \in \mathcal{E}_{\boldsymbol{x}} \\ 0 & \text{otherwise} \end{cases} \tag{24}$$

$$\operatorname{norm} \operatorname{topk}(\boldsymbol{s}) = \frac{\operatorname{topk}(\boldsymbol{s})}{\sum_i \operatorname{topk}(\boldsymbol{s})[i]} \tag{25}$$

### A.2 Measuring the Number of Active Channels in $\boldsymbol{u}$

In order to explore whether a ($\boldsymbol{k}_i$ - $\boldsymbol{v}_i$) sparsity-based approach is feasible, we measure the number of nonzero entries in the up-projected vector $\boldsymbol{u}$ in our baseline models (which, because of the ReLU activation function, is the same as the positive entries). We show the results of our 47M model in Fig. 1. Note that $d_{\text{ff}} = 2053$ (See Tab. 8) for the same model, which means that on average only 1-10% of the channels are active. We show the same analysis for the 262M model in Fig. 4. Interestingly, the counts remain the same, even though $d_{\text{ff}} = 4110$ for this model. The 41M parameter model on Enwik8 shows a stark difference in the distribution of the channels between layers; see Fig. 5. This suggests that the key factor determining the count distribution is the dataset, and the size of the model plays only a secondary role. Fortunately, the sparsity is very high for all models considered.

### A.3 More Details and Results on PKM

Our PKM (Sec. 3.2) is based on Lample et al. (2019) with the following basic modifications. First, we do not use batch normalization (BN). As Lample et al. (2019) shows that BN is only beneficial for models with a very large memory size, we remove it as it simplifies inference where the effective batch size varies over time. Also, we directly

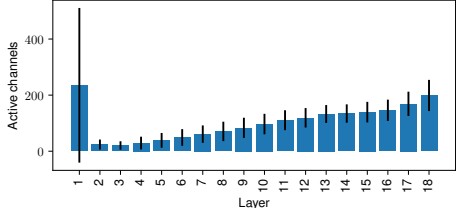

Figure 4: Number of active channels in $\boldsymbol{u}$ in our dense 262M parameter model on Wikitext-103. $d_{\text{ff}} = 4110$ for this model, so the sparsity is below $\sim 5\%$. Standard deviation over all tokens of the test and validation set.

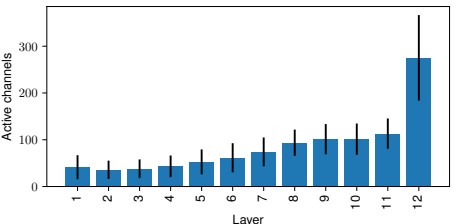

Figure 5: Number of active channels in $\boldsymbol{u}$ in our dense 41M parameter model on Enwik8. $d_{\text{ff}} = 2053$ for this model, thus the sparsity is below $\sim 15\%$. Standard deviation over all tokens of the test and validation set.

divide the input vectors into two sub-keys without an additional projection. Finally, unlike Lample et al. (2019), we use the same learning rate for all parts of the network.

In addition to the parameter-equal comparison of Sec. 6.2, there is another possibly "fair" way of setting the size of the PKM-based model: match the number of values (this would result in fewer parameters because of the key approximation), even though Elhage et al. (2022) suggest that the keys typically play a vital role, and reducing their capacity will cause a performance loss. See Tab. 6 for the corresponding results. Note that, for Enwik8 and Wikitext-103 small, the parameter-equal setting increases the number of sub-keys from 46 to 62 (2116 vs. 3844 values). This helps significantly.

### A.4 Further Analyses of Our $\sigma$-MoE

We also examine the best $(G, K)$ given a constant number $(G \cdot K)$ of active pairs $\boldsymbol{k}_i, \boldsymbol{v}_i$. In this setting, reducing $K$ by a factor of $m$ ($K' = \frac{K}{m}$) involves increasing $G$ ($G' = mG$), which, for a constant number of parameters, reduces $N_E$ to $N'_E = \frac{N_E}{m}$. The results can be seen in the 2nd block of Tab. 10. We find that a higher $K$ is beneficial. Given this, we ask the question how the selection distribution of the models with $K > 1$ is different from selecting the same experts together and acting as a larger

Table 6: The performance of the PKM model variants. Both value-count and parameter-matched variants are shown. Additionally, we show the effect of the initialization inspired by our unified view, which is marginal for PKMs.

| Variant | Setting | Nonlinearity | WT-S | WT-M | E8 |
|---|---|---|---|---|---|
| Dense Baseline | | ReLU | 11.81 | 9.46 | 1.08 |
| PKM | value-count | Softmax | 14.11 | 11.29 | 1.20 |
| PKM | value-count | ReLU | 13.32 | 10.16 | 1.12 |
| PKM | # total params. | Softmax | 13.96 | 11.10 | 1.16 |
| PKM | # total params. | ReLU | 12.77 | 9.98 | 1.11 |
| PKM + init | # total params. | ReLU | 12.75 | 9.96 | 1.11 |

expert. Are these models combining experts in more meaningful ways? To test this, we measure the distribution of experts that are used together on Wikitext-103 with our 47M MoE model with $K = 4$. The result can be seen in Fig. 6: the network combines experts in a rich way, further supporting the use of $K > 1$. Note that, it remains an open question whether such "compositions" may help the generalization and compositional behavior of the network (Fodor and Pylyshyn, 1988; Pagin and Westerståhl, 2010; Hupkes et al., 2020).

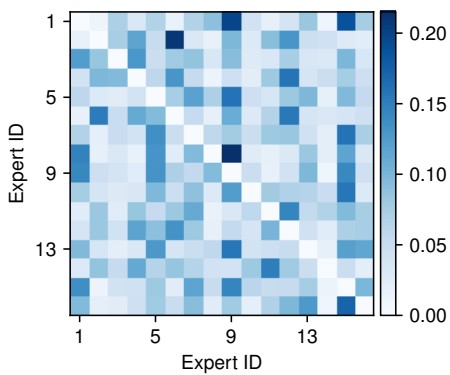

Figure 6: Expert co-occurrence in a $\sigma$-MoE model with $N_E = 16$ experts and $K = 4$. Each row shows the distribution of experts used together with the one corresponding to the row. Measured on the validation set of Wikitext-103 in the 3rd layer of our 47M $\sigma$-MoE model. The other layers and models behave qualitatively the same.

**Detailed Usage Count Analysis.** We show the relative proportion of experts selected for all layers in Fig. 7. For more details, please refer to Sec. 6.3.

### A.5 More on Resource Efficiency

For execution time and memory usage, both the dense MLP and the MoE layers are linear in $d_{\text{model}}$ (Fig. 9), the MLP is linear in $d_{\text{ff}}$, and MoE is linear in $G$ (Fig. 8) and $K$. For the same number of parameters (except for the selection network, which

is negligible), $d_{\text{model}} = G \cdot N_E$. However, both the memory usage and the execution time of the MoE are almost independent of $N_E$, except for a small linear factor due to the selection network (see Fig. 2). Figures 2, 8 and 9 show the actual measured execution time and memory usage on a RTX 3090 GPU.

Note that there is no significant difference in terms of speed and memory usage between different MoE variants given the same $d_{\text{model}}$, $G$, and $K$. This is because they only differ in the selection mechanism and regularization, and not in the way the experts are executed. Since all methods are configured to have the same number of parameters as the dense baselines, and $K$ experts are used in parallel, the factor of reduction in both FLOPs and memory usage is given by $\frac{K}{N_E}$. We show this factor for all models in Tab. 7.

### B Implementation details

We train all of our models for 100k steps with cosine learning rate decay, starting from the initial learning rate of 0.00025 and decaying to 0. We use the Adam optimizer (Kingma and Ba, 2015) with default PyTorch parameters (Paszke et al., 2019). We use gradient clipping with a max gradient norm of 0.25. We show the other hyperparameters of our dense models in Tab. 8. We train our models with an XL memory of the same size as the context size. However, following Dai et al. (2019), we evaluate the models using a longer memory. Unlike the hyperparameter-tuned memory sizes in Transformer XL, we use 4 times the context size (this approximates the size of the memory by Dai et al. (2019), while being simple).

The hyperparameters of the MoE models match those of their dense counterparts with the same number of parameters, except for the MoE-specific ones, which are shown in Tab. 9. $\delta$ denotes the expert dropout and $\gamma$ denotes the regularization

Table 7: The relative amount of FLOPs and memory used by the feedforward block of the MoE transformer compared to its dense counterpart. The same configurations are shown as in Tab. 10.

| Dataset | | | Wikitext 103 | Wikitext 103 | Wikitext 103 | Enwik8 |
| $d_{\text{model}}$ | | | 412 | 412 | 1024 | 512 |
| # params | | | 47M | 237M | 262M | 41M |
| | G | K | | | | |
| $\sigma$-MoE (ours) | 128 | 4 | 25.0% | 3.1% | 12.5% | 25.0% |
| standard dropout | 128 | 4 | 25.0% | 3.1% | 12.5% | 25.0% |
| softmax (after top-k) | 128 | 4 | 25.0% | 3.1% | 12.5% | 25.0% |
| softmax (before top-k) | 128 | 4 | 25.0% | 3.1% | 12.5% | 25.0% |
| standard init | 128 | 4 | 25.0% | 3.1% | 12.5% | 25.0% |
| no reg ($\gamma = 0, \delta = 0$) | 128 | 4 | 25.0% | 3.1% | 12.5% | 25.0% |
| $K = 8, G = 64$ | 64 | 8 | 25.0% | 3.1% | 12.5% | 25.0% |
| $K = 2, G = 256$ | 256 | 2 | 25.0% | 3.1% | 12.5% | 25.0% |
| $K = 1, G = 512$ | 512 | 1 | 25.0% | 3.1% | 12.5% | 25.0% |
| $N'_E = 2N_E, G = 64$ | 64 | 4 | 12.5% | 1.6% | - | 12.5% |
| $K = 1$ | 128 | 1 | 6.2% | 0.8% | - | 6.2% |
| $K = 2$ | 128 | 2 | 12.5% | 1.6% | - | 12.5% |
| $K = 8$ | 128 | 8 | 50.0% | 6.2% | - | 50.0% |
| Switch, $K = 1, G = 512$ | 512 | 1 | 25.0% | 3.1% | 12.5% | 25.0% |
| no dropout | 512 | 1 | 25.0% | 3.1% | 12.5% | 25.0% |
| $K = 4, G = 128$ | 128 | 4 | 25.0% | 3.1% | - | 25.0% |
| $K = 1, G = 128$ | 128 | 1 | 6.2% | 0.8% | - | 6.2% |
| no dropout | 128 | 1 | 6.2% | 0.8% | - | 6.2% |
| S-BASE | 128 | 4 | 25.0% | 3.1% | 12.5% | 25.0% |
| $K = 1, G = 512$ | 512 | 1 | 25.0% | 3.1% | 12.5% | 25.0% |

strength used for the loss $L$ (See Eq. 21). For the non-MoE layers, the same dropout is used as for the baselines. For Switch Transformers, we use $\gamma = 0.01$ with regularization of the form presented in Eq. 17, following Fedus et al. (2022). The other variants, including S-BASE, use the regularizer proposed by us (Eq. 21).

Our small PKM models use 46 subkeys resulting in $46^2 = 2116$ values for the $d_{\text{ff}}$-matched case and 62 subkeys (3844 values) for the parameter-matched case. The PKM equivalent of the 262M parameter model on Wikitext-103 has 64 subkeys (4096 values) for the $d_{\text{ff}}$-matched and 89 subkeys (7921 values) for the parameter-matched case. The PKM models do not use dropout in the PKM layers, and have 4 heads.

## B.1 A Few Words on the CUDA Kernel

We call the key operation for our MoE layers conditional vector-matrix multiplication, or CVMM, and we define it as follows. Given a batch of vectors, $\boldsymbol{V} \in \mathbb{R}^{N \times M}$, where $N$ is the batch size and $M$ is the number of channels, a set of $K$ matrices $\boldsymbol{M} \in \mathbb{R}^{K \times M \times L}$ and selection indices $\boldsymbol{S} \in$ $\{0, ..., K-1\}^N$, $\text{CVMM}(\boldsymbol{V}, \boldsymbol{S}, \boldsymbol{M}) \in \mathbb{R}^{N \times L}$ is:

$$\text{CVMM}(\boldsymbol{V}, \boldsymbol{S}, \boldsymbol{M})[n, l] = \qquad (26)$$
$$\sum_{m=0}^{M-1} \boldsymbol{V}[n, m] \boldsymbol{M}[\boldsymbol{S}[n], m, l]$$

Our CUDA kernel is based on the blog post developing a matrix multiplication kernel by Simon Boehm (https://siboehm.com/articles/22/CUDA-MMM). However, there are major differences: unlike standard matrix multiplication, in our case, different matrices could be used for different batch elements of the input. In order to be able to reuse matrices fetched from the global memory of the GPU, we first do a preprocessing step: we sort the selection indices, and obtain a reordering vector. This gives us an ordering of the input and output batch elements, such that the consecutive indices are multiplied by the same matrix with high probability. Fortunately, multiple channels have to be fetched/written out at once, so this reordering has minimal overhead. Our kernel has an additional grid dimension compared to standard matrix multiplication, iterating over the matrix index, $k \in \{0, ..., K-1\}$. We find that skipping matrices that do not have any corresponding inputs has minimal overhead. To avoid checking all elements of the reordering vector, we precompute their offsets.

Table 8: Hyperparameters of dense baselines and their MoE counterparts. For the MoE-specific hyperparameters, please refer to Tab. 9. "SetencePiece" tokenization is used for Wikitext-103, C4 and PES2O datasets, and "Character" for Enwik8.

| Tokenization | #params | $d_{\text{model}}$ | $d_{\text{ff}}$ | $n_{\text{layers}}$ | $n_{\text{heads}}$ | head size | context size | batch size | dropout | lr warmup |
|---|---|---|---|---|---|---|---|---|---|---|
| SentencePiece | 47M | 412 | 2053 | 16 | 10 | 41 | 256 | 64 | 0.1 | - |
| SentencePiece | 238M | 412 | 16480 | 16 | 10 | 41 | 256 | 64 | 0.1 | - |
| SentencePiece | 262M | 1024 | 4110 | 18 | 16 | 64 | 512 | 64 | 0.2 | 4000 |
| Character | 41M | 512 | 2053 | 12 | 8 | 64 | 512 | 32 | 0.1 | - |

Table 9: MoE-specific hyperparameters for different model variants. $\gamma$ denotes the scaler for the load balancing term in the loss and $\delta$ is the probability of the expert dropout. The standard, transformer-specific hyperparameters are the same as for the baselines. Please refer to Tab. 8. "SetencePiece" tokenization is used for Wikitext-103, C4 and PES2O datasets, and "Character" for Enwik8.

| Tokenization | #params | $d_{\text{model}}$ | $N_E$ | $G$ | $K$ | $\delta$ | $\gamma$ |
|---|---|---|---|---|---|---|---|
| SentencePiece | 47M | 412 | 16 | 128 | 4 | - | 0.001 |
| SentencePiece | 237M | 412 | 128 | 128 | 4 | 0.05 | 0.001 |
| SentencePiece | 262M | 1024 | 32 | 128 | 4 | 0.2 | 0.001 |
| Character | 41M | 512 | 16 | 128 | 4 | 0.05 | 0.0001 |

Our kernel uses shared memory and register caching; however, it does not use asynchronous loads, which makes it I/O bound. It also does not support tensor cores and mixed precision. The pre-processing step uses the radix sort from the CUB library. However, computing the offsets requires counting the number of vectors assigned to a single matrix. This information, as well as the offset, which is their sum, are freely available as sub-results that the radix sort computes anyways; however, we found no way of extracting it from the CUB implementation. We estimate that by implementing a more efficient preprocessing step, asynchronous loads, and tensor core support, our kernel can be further accelerated by a factor of two.

## B.2   Additional Results on MoEs

Additional results of different MoE variants with more model details are shown in Tab. 10. We repeat the entries from Tab. 4 for easier comparison.

Table 10: Detailed ablation results. WT-S* is obtained by naively scaling $N_E$ in WT-S. More details in Sec. 6.3. We do not evaluate all versions of the 262M Wikitext-103 model due to its long training time. However, we aim to include what we believe are the most interesting variants. $\gamma = 0$ means no regularization applied to the selection scores (See Eq. 21), $\delta = 0$ denotes no expert dropout.

| Variant | | | WT-S | WT-S* | WT-B | E8 |
|---|---|---|---|---|---|---|
| $d_{\text{model}}$ | | | 412 | 412 | 1024 | 512 |
| # params | | | 47M | 237M | 262M | 41M |
| | G | K | | | | |
| $\sigma$-MoE (ours) | 128 | 4 | 11.59 | 10.37 | 9.44 | 1.08 |
| standard dropout | 128 | 4 | 12.01 | 10.27 | 9.53 | 1.08 |
| softmax (after top-k) | 128 | 4 | 11.89 | 11.27 | 9.58 | 1.09 |
| softmax (before top-k) | 128 | 4 | 12.05 | 10.54 | 9.62 | 1.09 |
| standard init | 128 | 4 | 11.80 | 10.59 | 9.67 | 1.08 |
| no reg ($\gamma = 0, \delta = 0$) | 128 | 4 | 11.83 | 10.41 | 9.51 | 1.08 |
| $K = 8, G = 64$ | 64 | 8 | 11.63 | 10.30 | 9.58 | 1.08 |
| $K = 2, G = 256$ | 256 | 2 | 11.84 | 10.44 | 9.56 | 1.09 |
| $K = 1, G = 512$ | 512 | 1 | 11.90 | 10.83 | 9.58 | 1.09 |
| $N'_E = 2N_E, G = 64$ | 64 | 4 | 11.81 | 10.53 | - | 1.08 |
| $K = 1$ | 128 | 1 | 12.26 | 11.30 | - | 1.09 |
| $K = 2$ | 128 | 2 | 11.90 | 10.66 | - | 1.09 |
| $K = 8$ | 128 | 8 | 11.58 | 10.22 | - | 1.08 |
| Switch, $K = 1, G = 512$ | 512 | 1 | 12.27 | 11.24 | 9.68 | 1.08 |
| no dropout | 512 | 1 | 11.88 | 11.10 | 9.77 | 1.10 |
| $K = 4, G = 128$ | 128 | 4 | 12.05 | 11.37 | - | 1.10 |
| $K = 1, G = 128$ | 128 | 1 | 12.61 | 11.89 | - | 1.11 |
| no dropout | 128 | 1 | 12.35 | 11.78 | - | 1.10 |
| S-BASE, $K = 4, G = 128$ | 128 | 4 | 13.01 | 10.96 | 10.50 | 1.17 |
| $K = 1, G = 512$ | 512 | 1 | 12.32 | 11.31 | 9.77 | 1.32 |

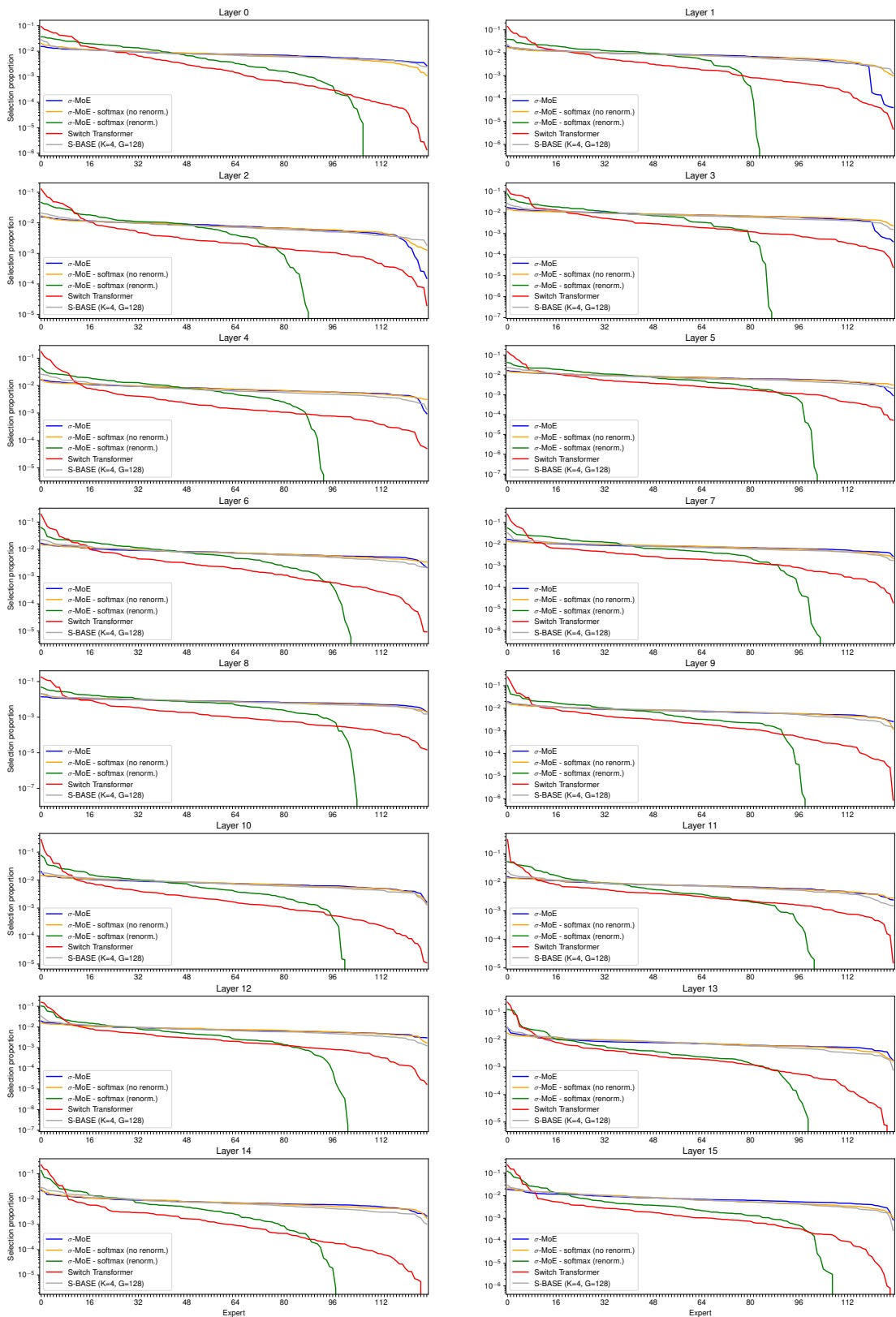

Figure 7: Total proportions of selection weights assigned to a given expert (x-axis) on the validation set of Wikitext-103 using the WT-S* models from Tab. 4. Experts are sorted by their popularity. All layers are shown ("Layer 5" is also shown in Fig. 3 in the main text). The models with poor performance can be distinguished easily (Switch Transformer and σ-MoE with a softmax and renormalization, "softmax (renom.)"). Their poor performance may be partially explained by expert collapse. In contrast, the fine performance differences between the rest of the models do not seem to be due to the expert collapse phenomenon.

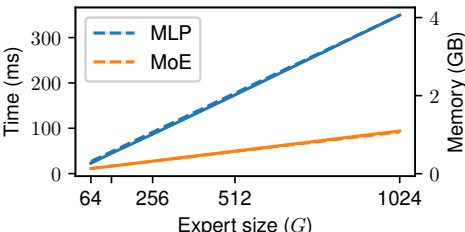

Figure 8: Measured execution time and memory usage of a forward-backward pass of a single MLP and MoE layer. $|\mathcal{B}| = 32768$, corresponding to the realistic scenario of a batch size 64 and sequence length 512, $d_{\text{model}} = 512$, $K = 4$, $N_E = 32$ and $d_{\text{ff}} = G \cdot N_E$. Full lines show the execution time, and dashed ones the memory consumption. Because they are both linear with similar slopes, they are almost indistinguishable. Even with our suboptimal CUDA kernel, the wall-clock time is faster starting from 16 experts.

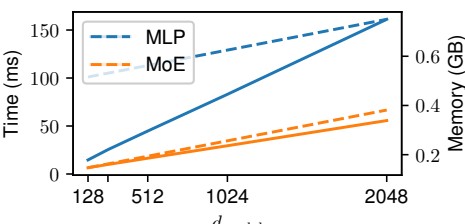

Figure 9: Measured execution time and memory usage of a forward-backward pass of a single MLP and MoE layer. $|\mathcal{B}| = 32768$, corresponding to the realistic scenario of a batch size 64 and sequence length 512, $K = 4$, $N_E = 32$, $G = 128$ and $d_{\text{ff}} = G \cdot N_E$. Full lines show the execution time, and dashed ones the memory consumption. Even with our suboptimal CUDA kernel, the wall-clock time is faster starting from 16 experts.