# OpenReview forum: "Approximating Two-Layer Feedforward Networks for Efficient Transformers"
_EMNLP/2023/Conference — EMNLP 2023 Findings_

### Official Review · Reviewer_FXRA · 2023-08-05

**Soundness:** 3

**Excitement:**

2: Mediocre: This paper makes marginal contributions (vs non-contemporaneous work), so I would rather not see it in the conference.

**Paper Topic And Main Contributions:**

The paper presents a comprehensive investigation of the use of sparse Mixtures of Experts (MoEs) and product-key-memories (PKMs) to improve the efficiency of any-scale-resource-efficient LMs in terms of computational and memory requirements. Evaluated under parameter-equivalent conditions, the empirical results underscore the competitive performance of MoEs compared to dense baselines.

**Questions For The Authors:**

The author emphasises that the \sigma MoE is applied to each MLP block of the model. What if the method is also applied in the usual way (e.g. once in every nth layer or even only in a single layer), how does the effectiveness change? Furthermore, including the number of layers applied, what is the main controllable factor in your approach that reduces FLOPs? In the \sigma n MoE initialisation? Why is the social care initialisation W_3 required? Does this effectiveness still validate on several downstream benchamark performance? What if the given reduced FLOPs are more than 50% (e.g. 75%) so that the Top-K activation function can be compared, can the proposed method outperform the Tok-K approach?

**Reasons To Accept:**

Clear comparisons to existing methods and evaluation direction. Upon analysis, the steps for the proposed method are straightforward. The effectiveness of the method is validated and ablation studies on different hyperparameter settings and components are provided.

**Reasons To Reject:**

Extending and combining the existing technique lacks novelty and, as pointed out in limitations, involves considerable sensitive engineering tricks. Lack of FLOPs information on other approaches' tables leads to unclear comparisons on how this approach is attractive in terms of FLOPs and bits/characters.

**Reproducibility:**

3: Could reproduce the results with some difficulty. The settings of parameters are underspecified or subjectively determined; the training/evaluation data are not widely available.

**Reviewer Confidence:**

2: Willing to defend my evaluation, but it is fairly likely that I missed some details, didn't understand some central points, or can't be sure about the novelty of the work.

---

> ### Author Rebuttal · Authors · 2023-08-28
>
> We thank the reviewer for the valuable time spent on reviewing our work.
>
> > *”Evaluated under parameter-equivalent conditions, the empirical results underscore the competitive performance of MoEs compared to dense baselines.”*
>
> Thank you very much for acknowledging this. In fact, evaluating MoEs under the “parameter-equivalent” condition is a unique and *novel* aspect of our study, challenging the common belief that MoEs underperform their dense counterparts under this condition. To the best of our knowledge, no prior work has shown MoEs are competitive under this setting.
>
> > *“Lack of FLOPs information on other approaches' tables leads to unclear comparisons on how this approach is attractive in terms of FLOPs and bits/characters”*
>
> Thank you for drawing our attention to this potentially confusing aspect.
> In fact, given the same hyperparameters ($d_{model}$, $G$ and $K$), all these MoE approaches have an almost identical number of flops and memory usage (namely, relative to their dense counterparts, both their FLOP and memory reduction factor are given by $\frac{G}{N_E}$). This is simply because they differ only in the way experts are selected and the selection mechanism only accounts for a negligible proportion of the total resources. This is the reason why we originally did not report FLOPS and the memory usage of other MoE baselines. That said, we fully agree with the reviewer that this was confusing as this explanation was missing in the current text. We will not only add the corresponding explanation, but also a new table displaying FLOPs/memory reduction compared to the dense model for all models/hyper-parameters configurations we evaluated:
>
>
> | Dataset | | | Wikitext 103 | Wikitext 103 | Wikitext 103 | Enwik8 |
> | --- | --- | --- | ---:   | ---:   | ---:   | ---:   |
> | $d_{model}$ | | | 412 | 412 | 1024 | 512 |
> | # params | | | 47M | 237M | 262M | 41M |
> | | G | K  |  |  |  |  |
> |  |  |  |    |    |    |    |
> | $\sigma$-MoE (ours) | 128 | 4 | 25.0\% | 3.1\% | 12.5\% | 25.0\% |
> | $~~~$ standard dropout | 128 | 4 | 25.0\% | 3.1\% | 12.5\% | 25.0\% |
> | $~~~$ softmax (after top-k) | 128 | 4 | 25.0\% | 3.1\% | 12.5\% | 25.0\% |
> | $~~~$ softmax (before top-k) | 128 | 4 | 25.0\% | 3.1\% | 12.5\% | 25.0\% |
> | $~~~$ standard init | 128 | 4 | 25.0\% | 3.1\% | 12.5\% | 25.0\% |
> | $~~~$ no reg ($\gamma=0,\delta=0$) | 128 | 4 | 25.0\% | 3.1\% | 12.5\% | 25.0\% |
> | $~~~$ $K=8, G=64$ | 64 | 8 | 25.0\% | 3.1\% | 12.5\% | 25.0\% |
> | $~~~$ $K=2, G=256$ | 256 | 2 | 25.0\% | 3.1\% | 12.5\% | 25.0\% |
> | $~~~$ $K=1, G=512$ | 512 | 1 | 25.0\% | 3.1\% | 12.5\% | 25.0\% |
> | $~~~$ $N_E' = 2N_E, G=64$ | 64 | 4 | 12.5\% | 1.6\% | -| 12.5\% |
> | $~~~$ $K=1$ | 128 | 1 | 6.2\% | 0.8\% | - | 6.2\% |
> | $~~~$ $K=2$ | 128 | 2 | 12.5\% | 1.6\% | - | 12.5\% |
> | $~~~$ $K=8$ | 128 | 8 | 50.0\% | 6.2\% | - | 50.0\% |
> |  |  |  |    |    |    |    |
> | Switch, $K=1, G=512$ | 512 | 1 | 25.0\% | 3.1\% | 12.5\% | 25.0\% |
> | $~~~$ no dropout | 512 | 1 | 25.0\% | 3.1\% | 12.5\% | 25.0\% |
> | $~~~$ $K=4, G=128$ | 128 | 4 | 25.0\% | 3.1\% | - | 25.0\% |
> | $~~~$ $K=1, G=128$ | 128 | 1 | 6.2\% | 0.8\% | - | 6.2\% |
> | $~~~$$~~~$ no dropout | 128 | 1 | 6.2\% | 0.8\% | - | 6.2\% |
> |  |  |  |    |    |    |    |
> | S-BASE | 128 | 4 | 25.0\% | 3.1\% | 12.5\% | 25.0\% |
> | $~~~$ $K=1, G=512$ | 512 | 1 | 25.0\% | 3.1\% | 12.5\% | 25.0\% |
>
>
>
> Additionally, here are some execution time and memory usage benchmarks for the different MoE models given different expert sizes and $d_\text{model}=512$, $K=4$, $N_{E}=32$ on a 2080Ti GPU (which empirically confirm our statement above about comparable time/memory usage among all the MoE methods):
>
> |  | 64 | 128 | 256 | 512 | 1024 |
> | --- |  ---: | ---: | ---: | ---: | ---: |
>  | $\sigma$-MoE | 17.8ms | 24.7ms | 43.3ms | 80.5ms | 155.5ms |
>  | Switch | 16.1ms | 24.9ms | 43.4ms | 80.5ms | 155.5ms |
>  | S-BASE | 16.6ms | 25.6ms | 44.1ms | 81.4ms | 156.7ms |
>
>  |  | 64 | 128 | 256 | 512 | 1024 |
> | --- |  ---: | ---: | ---: | ---: | ---: |
>  | $\sigma$-MoE | 135MB | 199MB | 327MB | 583MB | 1095MB |
>  | Switch | 135MB | 199MB | 327MB | 583MB | 1095MB |
>  | S-BASE | 135MB | 199MB | 327MB | 583MB | 1095MB |
>
> > *“What if the method is also applied in the usual way (e.g. once in every nth layer or even only in a single layer), how does the effectiveness change?”*
>
> Introducing more dense components will directly slow down the model. Since we show that our $\sigma$-MoE layer does not deteriorate the performance compared to the dense counterpart, using it only every n-th layer will only cause harm (we are not fully sure to understand what exactly the reviewer refers to as “effectiveness“).
>
>
> > *”Furthermore, including the number of layers applied, what is the main controllable factor in your approach that reduces FLOPs? “*
>
> In addition to the standard Transformer parameters such $d_{\text{model}}$, the size of the expert ($G$) and the number of active experts ($K$) also influence the FLOPs (we are not fully sure to understand the reviewer’s question, please let us know if this is effectively what the reviewer wished to know).
>
> > *”In the $\sigma$-MoE initialisation? Why is the social care initialisation $W_3$ required?”*
>
> The corresponding explanation is provided in Line 445. Essentially, each element of s, i.e, score $s\_k$ for expert selection, is a dot product between the input x and a row of $W\_3$. This initialization simply ensures that all rows of $W\_3$ have the same norm, such that scores initially only depend on the angles between x and the rows of $W\_3$ (otherwise there will be a bias towards rows with bigger norms to be selected, which may encourage collapsing).
>
> > *“Does this effectiveness still validate on several downstream benchamark performance?“*
>
> While it is not entirely to us what the reviewer refers to as *“effectiveness”*, the speed gain should remain effective for downstream tasks. Regarding the performance, we can not tell it based on our experiments. Here our scope is pure language modeling, and therefore, we evaluate them based on perplexity. The question of how the performance of MoE models in general transfers to downstream tasks compared to their dense counterpart is an interesting research direction for future work.
>
> > *”What if the given reduced FLOPs are more than 50\% (e.g. 75\%) so that the Top-K activation function can be compared, can the proposed method outperform the Tok-K approach?”*
>
> We are not fully sure to understand the reviewer’s question. Which model does *“if the given reduced FLOPs are more than 50\% (e.g. 75\%)“* refer to? We confirm that the FLOPs reduction is 75\% for the small MoE models and 87.5\% for the big ones (as shown in Table 3). The performance results for the top-K activation are already reported in Table 1. Comparing these to those in Table 4, we observe that the best sigma-MoE outperforms the top-K on the Wikitext-103/47M setting, (11.59 v. 11.68), underperforms it on Wikitext-103/262M (9.44 v. 9.26) and performs similarly on Enwik8 (1.08 v. 1.07). However, we would like to note that, unlike the MoE approaches, the acceleration obtainable by the top-K method alone is limited, as discussed in L199.
>
> We hope our response above brings clarifications to all the concerns raised by the reviewer.
> If you find our response useful, please consider increasing the scores. Thank you.

---

### Official Review · Reviewer_QrbU · 2023-08-09

**Soundness:** 4

**Excitement:**

3: Ambivalent: It has merits (e.g., it reports state-of-the-art results, the idea is nice), but there are key weaknesses (e.g., it describes incremental work), and it can significantly benefit from another round of revision. However, I won't object to accepting it if my co-reviewers champion it.

**Paper Topic And Main Contributions:**

The paper presents a novel Mixture-of-Experts (MoE) for Transformer models.

The paper starts with a review of existing MoE approaches, characterizing them as approximations of the 2-layer MLP of the transformer block, comparing it with other approximation schemes (top-k and Product-Key memories). After reviewing the different characteristics of these approaches, the paper proposes a novel MoE which uses a non-competitive expert selection function (sigmoid) followed by top-k selection, with a specific initialization scheme, expert dropout and a load-balancing regularization objective.

They evaluate the method on language modelling tasks, on standard datasets, and report that their approach provides comparable quality of parameter-equivalent dense models while being more compute-efficient, and also outperform other MoEs and approximation schemes. Ablations studies are also reported. The authors conclude that their MoE is beneficial even for small models.

**Reasons To Accept:**

- Interesting approach

- Compelling results

- The authors promise to release the code

**Reasons To Reject:**

- Some architectural choices are not entirely motivated

- Like all MoE approaches, this method is relatively complicated to implement compared to vanilla Transformers, requiring a custom CUDA kernel for efficient implementation.

**Reproducibility:**

3: Could reproduce the results with some difficulty. The settings of parameters are underspecified or subjectively determined; the training/evaluation data are not widely available.

**Reviewer Confidence:**

4: Quite sure. I tried to check the important points carefully. It's unlikely, though conceivable, that I missed something that should affect my ratings.

---

> ### Author Rebuttal · Authors · 2023-08-28
>
> We thank the reviewer for the valuable time spent on reviewing our work.
>
> > *“Some architectural choices are not entirely motivated”*
>
> We would have appreciated it a lot if the reviewer had specified *exactly* which “architectural choices” s/he is referring to. A general response is that all the introduced choices are motivated by the ultimate goal of making the approximate feedforward layers perform as good as the non-approximate one. We will be happy to clarify any explanations in our text that the reviewer still finds confusing.
>
> Also, some other reviewers (Reviewer d9Y9 in particular) also asked for clarification on certain design choices. Our response to them (we typically provide specific pointers to the part of the text that explains them) might also help the reviewer.
>
> > *”Like all MoE approaches, this method is relatively complicated to implement compared to vanilla Transformers, requiring a custom CUDA kernel for efficient implementation.”*
>
> As the reviewer wrote, this is true for all other MoE baselines or sparse acceleration methods in general. In fact, this is even true for many other well-known methods (e.g., even LSTM) before they became standard today. A custom CUDA kernel is ultimately required for any efficient implementation. Importantly (as the reviewer acknowledges), we will publicly release our kernel upon acceptance of our paper.
>
> Finally, we would like to emphasize one important aspect/novelty of our work which is omitted in the review. We demonstrate that MoEs are competitive to their dense counterparts even under the *parameter-equivalent condition*. This challenges the common belief that MoEs underperform their dense counterparts under this condition. To the best of our knowledge, no prior work has shown such a result (prior work only focused on the FLOP-equivalent comparison).
>
> We hope that our response above brings clarifications to the concerns raised by the reviewer.
> If the reviewer has any specific questions regarding the first point, we will be happy to answer them during the author-review discussion period.

---

### Official Review · Reviewer_JndA · 2023-08-10

**Typos Grammar Style And Presentation Improvements:** None
**Soundness:** 3

**Excitement:**

3: Ambivalent: It has merits (e.g., it reports state-of-the-art results, the idea is nice), but there are key weaknesses (e.g., it describes incremental work), and it can significantly benefit from another round of revision. However, I won't object to accepting it if my co-reviewers champion it.

**Missing References:**

None

**Paper Topic And Main Contributions:**

The paper studies an important problem, which is about reducing the compute and memory requirements of neural networks (NNs) without sacrificing performance. Specifically, the paper proposes using sparse Mixtures of Experts (MoEs) to create resource-efficient large language models (LMs). The paper presents a general framework that unifies various methods to approximate two-layer neural networks, including feedforward blocks of Transformers, and proposes methods to improve both MoEs and product-key memories (PKMs).

**Questions For The Authors:**

1. How does the approximation of the two-layer feedforward network affect the overall performance of the transformer model?

2. Have you conducted any experiments to evaluate the trade-off between efficiency and performance?

**Reasons To Accept:**

1. This paper proposes a new perspective on MoEs and a general framework that unifies various methods to approximate two-layer neural networks, which demonstrates the effectiveness of Mixtures of Experts (MoEs) in creating resource-efficient large language models (LMs).

2. This paper studies an important problem of reducing the computing and memory requirements of neural networks while maintaining performance. The paper proposes a novel method of using sparse Mixtures of Experts (MoEs) to create resource-efficient LMs and presents a general framework that unifies various methods to approximate two-layer neural networks. The paper's proposed methods to improve both MoEs and PKMs could also be useful for researchers and practitioners working on creating resource-efficient LMs.


**Reasons To Reject:**

1. The paper lacks a clear motivation for why approximating two-layer feedforward networks is important for efficient transformers. The authors should provide a more compelling argument for why this is a necessary step toward creating more efficient transformers.
2. The paper does not provide a thorough comparison of their proposed method with existing methods for reducing compute and memory requirements of neural networks. The authors should include a more comprehensive analysis of how their approach compares to other methods in terms of performance and efficiency.
3. The proposed method has not been subject to a comprehensive complexity analysis or a comparative analysis of training time. Figure 2 shows the execution time and memory usage of a forward-backward pass of a single MLP and MoE layer, however, the authors should provide more information on the computational complexity of their approach and how it compares to other MoE methods.
4. While the algorithmic design of the proposed method appears intuitive, the authors would benefit from a more detailed theoretical or analytical analysis of the existing content of the paper, which is currently lacking in detail. The authors should provide more information on the theoretical underpinnings of their approach and how it relates to existing research in the field.

**Reproducibility:**

4: Could mostly reproduce the results, but there may be some variation because of sample variance or minor variations in their interpretation of the protocol or method.

**Reviewer Confidence:**

2: Willing to defend my evaluation, but it is fairly likely that I missed some details, didn't understand some central points, or can't be sure about the novelty of the work.

---

> ### Author Rebuttal · Authors · 2023-08-28
>
> We thank the reviewer for the valuable time spent on reviewing our work.
>
> > *”1. The paper lacks a clear motivation for why approximating two-layer feedforward networks is important for efficient transformers. The authors should provide a more compelling argument for why this is a necessary step toward creating more efficient transformers.”*
>
> We’d like to draw the reviewer’s attention to the fact that every Transformer “layer” consists of one self-attention layer and one two-layer feedforward block. In typical large language models popular today, the feedforward block is configured to be very large, becoming an important computational bottleneck for efficient large transformer language models. This has motivated not only our work but also many prior works to accelerate it (for example, please see references for baseline models presented in Table 4).
>
> > *”2. The paper does not provide a thorough comparison of their proposed method with existing methods for reducing compute and memory requirements of neural networks. The authors should include a more comprehensive analysis of how their approach compares to other methods in terms of performance and efficiency.”*
> > *”3. Figure 2 shows the execution time and memory usage of a forward-backward pass of a single MLP and MoE layer, however, the authors should provide more information on the computational complexity of their approach and how it compares to other MoE methods.”* (We grouped Question 2 and 3 because they are directly related.)
>
> We first would like to point out that Tables 3 and 4 compare the performance of our models to different baselines, and Table 3 also compares their efficiency with their dense counterparts.
>
> That said, we agree with the reviewer that the comparison in terms of efficiency can still be potentially confusing in the current presentation.
> In fact, given the same hyperparameters ($d_{model}$, $G$ and $K$), all MoE approaches have an almost identical number of flops and memory usage (namely, relative to their dense counterparts, both their FLOP and memory reduction factor are given by $\frac{G}{N_E}$). This is simply because they differ only in the way experts are selected and the selection mechanism only accounts for a negligible proportion of the total resources. This is the reason why we originally did not report FLOPS and the memory usage of other MoE baselines.
>
> In the final version, we will not only add the corresponding explanation, but also a new table displaying FLOPs/memory reduction compared to the dense model for all models/hyper-parameters configurations we evaluated:
>
>
> | Dataset | | | Wikitext 103 | Wikitext 103 | Wikitext 103 | Enwik8 |
> | --- | --- | --- | ---:   | ---:   | ---:   | ---:   |
> | $d_{model}$ | | | 412 | 412 | 1024 | 512 |
> | # params | | | 47M | 237M | 262M | 41M |
> | | G | K  |  |  |  |  |
> |  |  |  |    |    |    |    |
> | $\sigma$-MoE (ours) | 128 | 4 | 25.0\% | 3.1\% | 12.5\% | 25.0\% |
> | $~~~$ standard dropout | 128 | 4 | 25.0\% | 3.1\% | 12.5\% | 25.0\% |
> | $~~~$ softmax (after top-k) | 128 | 4 | 25.0\% | 3.1\% | 12.5\% | 25.0\% |
> | $~~~$ softmax (before top-k) | 128 | 4 | 25.0\% | 3.1\% | 12.5\% | 25.0\% |
> | $~~~$ standard init | 128 | 4 | 25.0\% | 3.1\% | 12.5\% | 25.0\% |
> | $~~~$ no reg ($\gamma=0,\delta=0$) | 128 | 4 | 25.0\% | 3.1\% | 12.5\% | 25.0\% |
> | $~~~$ $K=8, G=64$ | 64 | 8 | 25.0\% | 3.1\% | 12.5\% | 25.0\% |
> | $~~~$ $K=2, G=256$ | 256 | 2 | 25.0\% | 3.1\% | 12.5\% | 25.0\% |
> | $~~~$ $K=1, G=512$ | 512 | 1 | 25.0\% | 3.1\% | 12.5\% | 25.0\% |
> | $~~~$ $N_E' = 2N_E, G=64$ | 64 | 4 | 12.5\% | 1.6\% | -| 12.5\% |
> | $~~~$ $K=1$ | 128 | 1 | 6.2\% | 0.8\% | - | 6.2\% |
> | $~~~$ $K=2$ | 128 | 2 | 12.5\% | 1.6\% | - | 12.5\% |
> | $~~~$ $K=8$ | 128 | 8 | 50.0\% | 6.2\% | - | 50.0\% |
> |  |  |  |    |    |    |    |
> | Switch, $K=1, G=512$ | 512 | 1 | 25.0\% | 3.1\% | 12.5\% | 25.0\% |
> | $~~~$ no dropout | 512 | 1 | 25.0\% | 3.1\% | 12.5\% | 25.0\% |
> | $~~~$ $K=4, G=128$ | 128 | 4 | 25.0\% | 3.1\% | - | 25.0\% |
> | $~~~$ $K=1, G=128$ | 128 | 1 | 6.2\% | 0.8\% | - | 6.2\% |
> | $~~~$$~~~$ no dropout | 128 | 1 | 6.2\% | 0.8\% | - | 6.2\% |
> |  |  |  |    |    |    |    |
> | S-BASE | 128 | 4 | 25.0\% | 3.1\% | 12.5\% | 25.0\% |
> | $~~~$ $K=1, G=512$ | 512 | 1 | 25.0\% | 3.1\% | 12.5\% | 25.0\% |
>
>
>
> Additionally, here are some execution time and memory usage benchmarks for the different MoE models given different expert sizes and $d_\text{model}=512$, $K=4$, $N_{E}=32$ on a 2080Ti GPU (which empirically confirm our statement above about comparable time/memory usage among all the MoE methods):
>
> |  | 64 | 128 | 256 | 512 | 1024 |
> | --- |  ---: | ---: | ---: | ---: | ---: |
>  | $\sigma$-MoE | 17.8ms | 24.7ms | 43.3ms | 80.5ms | 155.5ms |
>  | Switch | 16.1ms | 24.9ms | 43.4ms | 80.5ms | 155.5ms |
>  | S-BASE | 16.6ms | 25.6ms | 44.1ms | 81.4ms | 156.7ms |
>
>  |  | 64 | 128 | 256 | 512 | 1024 |
> | --- |  ---: | ---: | ---: | ---: | ---: |
>  | $\sigma$-MoE | 135MB | 199MB | 327MB | 583MB | 1095MB |
>  | Switch | 135MB | 199MB | 327MB | 583MB | 1095MB |
>  | S-BASE | 135MB | 199MB | 327MB | 583MB | 1095MB |
>
> We will add these tables to the final version.
> Thank you for drawing our attention to this potentially confusing aspect.
>
> > *“4. While the algorithmic design of the proposed method appears intuitive, the authors would benefit from a more detailed theoretical or analytical analysis of the existing content of the paper, which is currently lacking in detail. The authors should provide more information on the theoretical underpinnings of their approach and how it relates to existing research in the field.”*
>
> We are not fully sure to understand exactly what kind of *“more detailed theoretical or analytical analysis of the existing content of the paper”* the reviewer is requesting. Regarding *“theoretical underpinnings of their approach and how it relates to existing research in the field”*, we’d like to claim that our current Section 2, 3 and 4 provide a rather comprehensive summary of conceptual comparisons between different approaches. Sec 2 starts with a general framework (theoretical view) of approximate two-layer feedforward networks. Sec 3 then introduces top-K activations (an essential component of any MoEs) and connects PKMs and MoEs. Finally, Sec 4 reviews and conceptually compares existing MoE methods, before describing our method in Sec 5. If the reviewer still thinks that this is not enough, please specifically tell us what is missing. Thank you.
>
> > *“How does the approximation of the two-layer feedforward network affect the overall performance of the transformer model?”*
>
> Essentially, all our experimental results (Tables 1, 2, 3, 4) show *“how approximation of the two-layer feedforward network affects the overall performance of the transformer model”*. These tables evaluate *different* types of approximate methods for the two-layer feedforward networks.
>
> > *”Have you conducted any experiments to evaluate the trade-off between efficiency and performance?”*
>
> Yes, Table 3 compares both FLOPs and the performance (as measured by perplexity). Using our sigma-MoE, we obtain 75\% - 87.5\% speed up in FLOPs without losing performance. We also reported additional numbers related to this in the new tables above.
>
> We hope our response brings clarifications and improves the reviewer’s view on our contributions.

---

### Official Review · Reviewer_d9Y9 · 2023-08-14

**Soundness:** 3

**Excitement:**

3: Ambivalent: It has merits (e.g., it reports state-of-the-art results, the idea is nice), but there are key weaknesses (e.g., it describes incremental work), and it can significantly benefit from another round of revision. However, I won't object to accepting it if my co-reviewers champion it.

**Paper Topic And Main Contributions:**

This paper explores diverse techniques for approximating two-layer neural networks (NNs). To begin with, the authors introduce a comprehensive framework that unifies the Top-K activation function, Mixtures of Experts (MoEs), and product-key memories (PKMs). By thoroughly analyzing their approach, they subsequently present enhancements for both MoEs and PKMs. The empirical investigations reveal that the proposed MoEs perform competitively when compared to the dense Transformer-XL, underscoring the applicability of MoEs to language models of varying scales.

**Questions For The Authors:**

1. It would be valuable to include additional experiments on diverse text datasets such as PTB and C4. Expanding the experimental evaluation beyond Enwik8 and WikiText-103 can provide a more comprehensive understanding of the proposed approach's performance across various text domains and scales.
2. Could you provide further clarification and elaboration on the design principles underlying the $\sigma$-MoE model?

**Reasons To Accept:**

1. The exploration of approximating two-layer feedforward networks (FFNs) is both novel and captivating. Notably, the authors provide a comprehensive perspective that encompasses various methods, and they also conduct a comprehensive comparison of the most notable MoE variants.
2. The motivation behind the paper is distinctly articulated and substantiated. The seamless progression from motivation to theoretical analysis, and ultimately to the proposal of the method, is presented in a coherent and natural manner.
3. The proposed approach is effectively supported by empirical experiments, demonstrating its ability to attain performance on par with dense networks.

**Reasons To Reject:**

1. The proposed $\sigma$-MoE framework requires further justification. The specific design approach for $W_3$ and the uniform initialization of weight matrices are notable contributions. The paper should delve into the rationale behind these design choices and their benefits, particularly in terms of their impact on the performance of the MoE model.

2. The empirical investigations provided appear somewhat limited in scope. Given that all experiments are exclusively conducted on WikiText-103 and Enwik8, which share similar data distributions, it would be prudent to expand the experimental scope to include other datasets. This would provide additional support for the performance claims of the $\sigma$-MoE model.

3. While the paper introduces novel elements through the MoE variant design, the novelty level might be constrained. To enhance the clarity of the novelty introduced beyond the MoE variant design, it's advisable to provide further elaboration and illustration.



**Reproducibility:**

4: Could mostly reproduce the results, but there may be some variation because of sample variance or minor variations in their interpretation of the protocol or method.

**Reviewer Confidence:**

3: Pretty sure, but there's a chance I missed something. Although I have a good feel for this area in general, I did not carefully check the paper's details, e.g., the math, experimental design, or novelty.

---

> ### Author Rebuttal · Authors · 2023-08-28
>
> We thank the reviewer for the valuable time spent on reviewing our work.
>
> > *“The empirical investigations reveal that the proposed MoEs perform competitively when compared to the dense Transformer-XL, underscoring the applicability of MoEs to language models of varying scales.”*
>
> Thank you very much for acknowledging this. We would like to emphasize that this is under the *parameter-equivalent condition*, which is a unique and *novel* aspect of our study, challenging the common belief that MoEs underperform their dense counterparts under this condition. To the best of our knowledge, no prior work has shown that MoEs are competitive in this setting.
>
> > *“1. The proposed sigma-MoE framework requires further justification. ... The paper should delve into the rationale behind these design choices and their benefits, particularly in terms of their impact on the performance of the MoE model.”*
> > (Question 2) *”Could you provide further clarification and elaboration on the design principles underlying the sigma-MoE model?”*
>
> We are sorry for the confusion. We will emphasize these points better in the final version of our paper. The $\sigma$-MoE consists of the following key architectural differences compared to previous work: non-competitive selection function (sigmoid), global entropy regularization, optional expert dropout, and the initialization of the selection mechanism. Here is a short summary of the rationale for each of them:
>
> - Sigmoid activation function: Section 3 highlights the similarity between the MoEs and the top-k approximation of the feedforward layers; we observe that one of the key differences
> between them is the use of a competitive (softmax in MoEs) vs. non-competitive activation function (in standard FF layers).
> In Sec 6.1 we empirically show that the top-k approximation of the FF layers performs well.
> This success has motivated us to also use a non-competitive activation function (sigmoid) in our MoE.
> This explanation is given in line 403.
> - Global entropy regularization: We found that the existing regularization methods were complex and somewhat arbitrary (L451). Our goal was to have each of the experts used for some processing step, with minimal restrictions. A natural way to achieve this is to regularize the entropy of the distribution of the selection scores (computed within the batch, in practice). This results in Eqs. 20 and 21.
> - Expert dropout (L465-476): Our primary goal was to increase expert utilization.
> Randomly dropping experts help us prevent the collapsing phenomena where only a few (or single) experts are consistently used and trained.
> - Initialization of the selection mechanism: The rationale is briefly described in line 445. We normalize the length of the row vectors of the projection matrix. This is helpful because, given that each row is responsible for computing the score of the corresponding expert by taking the dot product with the input, if any of the rows had a smaller norm than the others, the corresponding expert would be selected more rarely, which would encourage the collapsing (assuming a uniform distribution of the input vectors).
>
> Following the reviewer’s suggestion:
>
> > *”3. To enhance the clarity of the novelty introduced beyond the MoE variant design, it's advisable to provide further elaboration and illustration.”*
>
> We will add a figure summarizing these modifications in the final version. If the reviewer has any better/concrete suggestions, we’ll be also happy to take them.
>
> > *“2. it would be prudent to expand the experimental scope to include other datasets.”*
> > (Question 1) *”It would be valuable to include additional experiments on diverse text datasets such as PTB and C4.”*
>
> We would like to thank the reviewer for this suggestion. We conducted extra experiments using the same models on C4 and the newly proposed peS2o dataset (https://github.com/allenai/peS2o, a collection of cleaned, filtered text from academic papers). We chose peS2o instead of PTB because PTB is tiny and is known to be all about "regularization" (see for example Merity et al. 2017: Regularizing and Optimizing LSTM Language Models). We used identical hyperparameters as in the Wikitext 103 experiments in our paper, including the same number of iterations. This means that we cannot do a full epoch on the training data, which would be prohibitively expensive with the compute budget we have. Instead, we train it for the same number of tokens as for Wikitext 103. The results are as follows:
>
> |Dataset | | | C4 | C4 | peS2o | peS2o |
> | --- | --- | --- | ---:   | ---:   | ---:   | ---:   |
> | $d_{model}$ | | | 412 | 1024 | 412 | 1024 |
> | # params | | | 47M |  262M |  47M |  262M |
> | | G | K  |  |  |  |  |
> |  |  |  |    |    |    |    |
> | Dense | 128 | 1 | 23.76 | 17.79 | 14.34 | 10.91 |
> | $\sigma$-MoE (ours) | 128 | 4 | 23.25 | 17.46 | 14.12 | 10.91 |
> | Switch, $K=1, G=512$ | 512 | 1 | 24.47 | 18.29 | 14.74 | 11.56 |
> | S-BASE | 128 | 4 | 35.48 | 18.53 | 16.61 | 11.72 |
>
> This confirms the results we reported in the paper on two more datasets.
> We hope that this helps to clear the reviewer's doubts about the generality of our method.

---

### Meta-Review · Area_Chair_2L7C · 2023-09-19

**Recommendation:** 3

**Metareview:**

The manuscript underwent a comprehensive evaluation by four reviewers, revealing both strengths and weaknesses in its content.

Pros

1. Novelty and Contribution: The reviewers largely concur that the paper provides a novel investigation into the approximation of two-layer feedforward networks.
2. Empirical Robustness: The empirical outcomes are broadly considered to be compelling, particularly highlighting the MoEs' competitive edge over dense Transformer architectures.
3. Community Relevance: By focusing on diminishing computational and memory overhead without compromising model efficacy, the paper holds significant appeal for both researchers and practitioners in the field.


Cons

1. Lack of Rigorous Justification: The design choices within the MoE framework warrant further elucidation. The reviewers note that these choices are not sufficiently justified, especially regarding their impact on model performance, and the authors were unable to clarify their motivations convincingly in their rebuttal.
2. Need for Novelty Elaboration: Though the paper incorporates novel components, some reviewers question the extent of this novelty and advocate for a more detailed exposition to distinguish the proposed method from existing MoEs.
3. Comparative Evaluation: Reviewers express the need for a more systematic comparison with established methodologies, focusing on aspects such as computational complexity, performance metrics, and overall efficiency. The rebuttal partially resolved this issue.
4. Theoretical Depth: A deeper theoretical or analytical discourse is recommended by the reviewers to both validate the design decisions and to situate the work more solidly within the existing body of literature.

---

### Decision · Program_Chairs · 2023-10-07

**Decision:**

Accept-Findings

**Comment:**

The manuscript underwent a comprehensive evaluation by four reviewers, revealing both strengths and weaknesses in its content.

Pros

1. Novelty and Contribution: The reviewers largely concur that the paper provides a novel investigation into the approximation of two-layer feedforward networks.
2. Empirical Robustness: The empirical outcomes are broadly considered to be compelling, particularly highlighting the MoEs' competitive edge over dense Transformer architectures.
3. Community Relevance: By focusing on diminishing computational and memory overhead without compromising model efficacy, the paper holds significant appeal for both researchers and practitioners in the field.


Cons

1. Lack of Rigorous Justification: The design choices within the MoE framework warrant further elucidation. The reviewers note that these choices are not sufficiently justified, especially regarding their impact on model performance, and the authors were unable to clarify their motivations convincingly in their rebuttal.
2. Need for Novelty Elaboration: Though the paper incorporates novel components, some reviewers question the extent of this novelty and advocate for a more detailed exposition to distinguish the proposed method from existing MoEs.
3. Comparative Evaluation: Reviewers express the need for a more systematic comparison with established methodologies, focusing on aspects such as computational complexity, performance metrics, and overall efficiency. The rebuttal partially resolved this issue.
4. Theoretical Depth: A deeper theoretical or analytical discourse is recommended by the reviewers to both validate the design decisions and to situate the work more solidly within the existing body of literature.